# An LES Model for Wind Farm-Induced Atmospheric Gravity Wave Effects Inside Conventionally Neutral Boundary Layers.

Sebastiano Stipa[1], Mehtab Ahmed Khan[2], Dries Allaerts[2], and Joshua Brinkerhoff[1]

[1]School of Engineering, University of British Columbia–Okanagan, Kelowna, Canada
[2]Aerospace Engineering, TU Delft, Delft, the Netherlands

**Correspondence:** Sebastiano Stipa (sebstipa@mail.ubc.ca)

**Abstract.** The interaction of large wind farm clusters with the thermally-stratified atmosphere has emerged as an important physical process that impacts the productivity of wind farms. Under stable conditions, this interaction triggers atmospheric gravity waves (AGWs) in the free atmosphere due to the vertical displacement of the atmospheric boundary layer (ABL) by the wind farm. AGWs induce horizontal pressure gradients within the ABL that alter the wind speed distribution within the farm, influencing both wind farm power generation and wake development. Additional factors, such as the growth of an internal boundary layer originating from the wind farm entrance and increased turbulence due to the wind turbines, further contribute to wake evolution. Recent studies have highlighted the considerable computational cost associated with simulating gravity wave effects within large eddy simulations (LES), as a significant portion of the free atmosphere must be resolved due to the large vertical spatial scales involved. Additionally, specialized boundary conditions are required to prevent wave reflections from contaminating the solution. In this study, we introduce a novel methodology to model the effects of AGWs without extending the LES computational domain into the free atmosphere. The proposed approach addresses the wave reflection problem inherently, eliminating the need for these specialized boundary conditions. We utilize the recently developed multi-scale coupled (MSC) model of Stipa et al. (2024b) to estimate the vertical ABL displacement triggered by the wind farm, and apply the deformation to the domain of an LES that extends only to the inversion layer. The accuracy in predicting the AGW induced pressure gradients is equivalent to the MSC model. The AGW modeling technique is verified for two distinct free atmosphere stability conditions by comparing it to the traditional approach in which AGWs are fully resolved using a domain that extends several kilometers into the free atmosphere. The proposed approach accurately captures AGW effects on the row-averaged thrust and power distribution of wind farms while demanding 12.7% of the computational resources needed for traditional methods. Moreover, when conventionally neutral boundary layers are studied there is no longer a need for solving the potential temperature equation, as stability is neutral within the boundary layer. The developed approach is subsequently used to compare global blockage and pressure disturbances obtained from the simulated cases against a solution characterized by zero boundary layer displacement, which represents the limiting case of very strong stratification above the boundary layer. This approximation, sometimes referred to as the "rigid lid", is compared against the full AGW solution using the MSC model. This is done for different values of inversion strength and free atmosphere lapse rate, evaluating the ability of the "rigid lid" to predict blockage, wake effects and overall wind farm performance.

# 1 Introduction

Wind farms, especially those situated offshore, are increasing both in number and size, interacting with the atmosphere well beyond their physical boundaries. Such interactions play an important role both in the evolution of cluster wakes and on the amount of flow deceleration experienced upstream, also known as blockage. On the one hand, wind farm wake recovery is greatly influenced by surface stability within the atmospheric boundary layer (ABL). On the other hand, the stably stratified free atmosphere leads to the generation of atmospheric gravity waves (AGWs) when mean flow streamlines are vertically perturbed by the wind farm. These waves exist in the form of interface waves within the capping inversion layer and internal gravity waves aloft, introducing a pressure feedback mechanism at the wind farm scale that ultimately impacts the flow dynamics inside the ABL (Smith, 2010). In contrast to terrain-generated gravity waves (Smith, 1980, 2007; Teixeira, 2014), gravity waves triggered by wind farms yield lower pressure and velocity perturbations inside the ABL when these are compared against turbulent fluctuations. Moreover, similarly to mountain waves, wind farm-induced AGWs are characterized by horizontal and vertical wavelengths of $\mathcal{O}(10)$ km, depending on the wind farm length, specific potential temperature structure and geostrophic wind. These aspects make AGW observation and experimental measurement difficult to achieve. Because of such complexities, wind farm-induced AGWs have mainly been studied by means of high-fidelity models such as large eddy simulations (LES) or using linear gravity-wave theory (Nappo, 2012; Lin, 2007). LESs of AGWs suffer from high computational cost and AGW reflections at the numerical boundaries. The aim of this study is to overcome these difficulties by using a reduced order model based on linear theory to construct an LES methodology that significantly simplifies the inclusion of AGW effects within the ABL flow.

Using a two-dimensional numerical model, the impact of gravity waves on the flow around hills and complex terrain was first studied by Klemp and Lilly (1978), who addressed the problem of wave reflection from the upper boundary. Later, wind farm-induced AGWs were investigated using LES for conventionally neutral boundary layers (CNBLs) by Allaerts and Meyers (2017), Lanzilao and Meyers (2022), Stipa et al. (2024b) and Lanzilao and Meyers (2024), among others. CNBLs are characterized by a neutral stratification within the ABL, followed by a positive potential temperature jump $\Delta\theta$ across the inversion layer and a linear lapse rate $\gamma$ in the free atmosphere aloft. These studies showed that the presence of AGWs has two main implications, namely an adverse pressure gradient upwind of the wind farm, which is responsible for global blockage, and a favorable pressure gradient inside the wind farm that is beneficial for wake recovery. Moreover, Centurelli et al. (2021) and Maas (2023) showed that LES results strongly differ from reduced-order wake models when thermal stratification is considered. To assess the impact of inversion height, strength and lapse rate on wind farm blockage and wind farm efficiency in general, Lanzilao and Meyers (2024) conducted an LES parametric study, concluding that the overall effect of AGWs on wind farm performance can be either beneficial or detrimental depending on the specific structure of the potential temperature profile. This result highlights the importance of including AGWs when modeling wind farms, both in high-fidelity and low-fidelity models.

An important aspect that emerges from the above studies, highlighted by Lanzilao and Meyers (2024), is that LESs of wind farms including AGWs are challenging. Firstly, they are rendered computationally intense by the domain size required to spatially resolve AGWs. Furthermore, special boundary conditions should be used to damp out AGWs before they reach the

domain boundaries and reflect, contaminating the solution. To overcome this issue, different approaches have been proposed so far. Béland and Warn (1975) and Bennett (1976) constructed transient radiation boundary conditions using the Laplace transform, but these require storing the entire flow history at each reflecting boundary. Klemp and Durran (1983) overcame this limitation by deriving a radiation condition for the top boundary that is local in time, using the linear, hydrostatic, Boussinesq equations. The authors also showed that low AGW reflectivity is still observed when these hypotheses are not strictly met,

a result that was later confirmed by Lanzilao and Meyers (2023). Another approach that avoids AGW reflections at the top boundary is the so-called Rayleigh damping layer (Klemp and Lilly, 1978). This is a region of the domain where the momentum equation features an extra source term, proportional to the difference between the perturbed and unperturbed ABL states. In theory, this eliminates AGWs before they can reach the boundary but the proportionality coefficient, which increases with height, should be properly tuned. Reflections may still be observed both when damping is too strong or when it is too weak.

In the first case, the Rayleigh damping region behaves as a physical boundary, while in the latter the damping is insufficient to cancel out perturbations before they reach the physical boundary.

    Some guidelines on how to choose the Rayleigh damping parameters have been provided by Lanzilao and Meyers (2023) and Klemp and Lilly (1978). Moreover, many other studies (see Allaerts and Meyers, 2017, 2018, among others) agree that the Rayleigh damping layer located at the top should be larger than the expected vertical wavelength of the AGWs, estimated

as $\lambda_z = 2\pi G/N$, where $N$ is the Brunt-Väisälä frequency and $G$ is the geostrophic wind. Similarly, the vertical extent of free atmosphere included within the simulation domain must allow at least one vertical wavelength to be resolved.

    Non-reflecting boundaries are also required in the horizontal directions, but their implementation is complicated by the fact that they should not alter the incoming ABL turbulence. In this case, two options are possible. The first is the so-called fringe region technique (Inoue et al., 2014), which is essentially a Rayleigh damping layer where the unperturbed state used

to compute the momentum source term is local in both time and space. This requires a separate simulation of the unperturbed flow, referred to as the concurrent precursor, to run concurrently with the main simulation, i.e. the successor. This ensures that a time- and spatially-resolved reference turbulent flow is available within the fringe region to compute the damping source at each iteration. As the concurrent precursor naturally contains the incoming turbulence, this technique eliminates AGW while simultaneously prescribing the unperturbed turbulent inflow to the successor simulation. Similarly to the Rayleigh damping

layer, the fringe region requires *ad-hoc* tuning of the proportionality coefficient that controls the amount of damping, which is usually accomplished by trial and error, further raising computational costs. Notably, Lanzilao and Meyers (2023) observed that the fringe region itself may trigger spurious gravity waves while attempting to restore the unperturbed state, requiring an additional layer in which horizontal advection of vertical momentum is suppressed to prevent these spurious perturbations to be transported downstream. Another possibility to avoid wave reflections at the inlet and outlet boundaries is to use Rayleigh

damping regions above the boundary layer (Mehtab Ahmed Khan, personal communication 2023), so that turbulence remains unaffected below. However, this technique requires to accurately chose the horizontal unperturbed flow in the free atmosphere and poses issues when a capping inversion layer is present.

    Regarding the description of AGW by means of reduced-order models, this was first achieved by Smith (1980, 2007) for the flow around terrain features, in what is referred to as the two-layer model (2LM). The 2LM exploits the linear theory

for interacting gravity waves and boundary layers, and was later extended to wind farms immersed in CNBLs (Smith, 2010). Building on his work, Allaerts and Meyers (2019) developed the three-layer model (3LM), a substantial improvement of the 2LM characterized by extra features such as the Coriolis force, the additional wind farm layer that relaxes Smith's homogeneous vertical mixing assumption, and the wind farm/gravity wave coupling mechanism. Although the 3LM was the first study to incorporate AGW effects into predictions of wind farm power losses, it lacked a local coupling between the mesoscale and

turbine scales, failing to address the effects of gravity wave induced pressure gradients inside the wind farm and in the wake. Recently, Devesse et al. (2023) and Stipa et al. (2024b) proposed new localized coupling strategies between the 3LM and conventional wake models that capture all features of the wind farm interaction with AGWs under CNBLs. The latter, referred to as the multi-scale coupled model (MSC), is characterized by a lower computational cost and its formulation is independent of the adopted wake model.

When thermal stratification above the ABL is very strong, lapse rate and inversion strength lose importance and the background pressure gradient is mainly determined by the boundary layer height. In this case, the flow cannot be perturbed vertically because thermal stratification acts as a lid located at the ABL top. Such idealized limiting case is commonly referred to as the rigid lid approximation (Smith, 2023). As the lid imposes zero mean vertical mass flux, the solution is characterized by a harmonic perturbation pressure that renders the mean flow horizontally divergence-free, with maximum and minimum pressure

at the wind farm start and exit, respectively. The rigid lid approximation maintains some properties of the full gravity wave solution, such as the presence of global blockage and flow acceleration within the wind farm. This, combined to its inherently simpler formulation than the full AGW solution, makes the rigid lid approximation worth investigating for its potential use in engineering parametrizations.

The methodology proposed in this study allows to model AGW effects within a wind farm LES while eliminating the

computational burden associated with resolving internal and interfacial waves above the ABL. In fact, while LES is used below the inversion layer, AGWs in the free atmosphere and within the inversion layer are modeled through the MSC model (Stipa et al., 2024b) using the vertical ABL displacement as the coupling variable. As a consequence, the developed approach only requires a vertical domain size that is equal to the height of the inversion layer, assumed to coincide with the ABL height. Moreover, no damping regions are needed as the large-scale pressure gradients produced by AGWs are modeled without

resolving the actual waves. Finally, when dealing with CNBLs, the flow is neutral within the boundary layer and the solution of a potential temperature equation can be omitted. Although the proposed method could be applied in principle to internally stable ABLs by solving for the potential temperature equation, the accuracy of the MSC model — to which the LES solution depends — has not been tested in this condition yet. Hence, this manuscript solely focuses on CNBLs, leaving internal ABL stability as an object for future investigation.

The present paper is organized as follows. Section 2 describes the proposed LES methodology, pointing out its differences with respect to the conventional approach used to simulate the wind farm/gravity wave interaction. Section 3 describes the set-up of the LES cases used to verify the proposed methodology. Model verification is presented in Section 4, together with an analysis regarding the implications of using the rigid lid approximation. Finally, Section 5 highlights the conclusions of the present study.

## 2  LES Methodology

For the LES simulations presented in this paper, we use the open-source finite volume code TOSCA (Toolbox fOr Stratified Convective Atmospheres) developed at the University of British Columbia and extensively validated in Stipa et al. (2024a). In order to distinguish between AGW-resolved simulations and the proposed approach, we first describe, in Section 2.1, the characteristics of a simulation that naturally resolves AGWs and their effects within the boundary layer. Then, in Section 2.2, we present the proposed modeling strategy with guidelines on its application. Here, only the turbulent part of the CNBL is included in the LES domain, while the steady-state solution in the free atmosphere is obtained from the MSC model of Stipa et al. (2024b).

### 2.1  AGW-Resolved Approach

Large wind farms may trigger interface waves when an inversion layer is present as well as internal waves in the stably stratified free atmosphere (Lin, 2007; Nappo, 2012) by steadily perturbing the boundary layer height vertically. Although the extent of such perturbation also depends on the level of stability experienced inside the boundary layer, the present study only focuses on CNBLs. Governing equations correspond to mass and momentum conservation for an incompressible flow with Coriolis forces and Boussinesq approximation for the buoyancy term. The latter is calculated using the modified density $\rho_k$, evaluated by solving a transport equation for the potential temperature. The exact form of the equations implemented in TOSCA and used in the present study is reported in Stipa et al. (2024a).

When simulating AGWs in an LES framework, the simulation domain should extend to one or more wavelengths in each direction (Klemp and Lilly, 1978; Allaerts and Meyers, 2019; Stipa et al., 2024b; Lanzilao and Meyers, 2024). Under conditions that are representative of normal wind farm operation, i.e. lapse rates ranging between $1-10$ K km$^{-1}$ and geostrophic winds of $5-20$ m s$^{-1}$, $\lambda_z$ can be between $2-20$ km. In addition, waves inherently reflect if they do not decay before reaching boundaries, either requiring the computational domain to span several wavelengths or the use of damping regions where AGWs are artificially damped. With reference to the different boundary conditions listed in Section 1 at the domain top, the Rayleigh damping region represents the best solution in terms of wave reflectivity (Lanzilao and Meyers, 2023). This is prescribed by applying a source term in the vertical momentum balance calculated as

$$s^r(\boldsymbol{x}) = \nu_r(z)\left[\overline{w} - w(\boldsymbol{x},t)\right], \tag{1}$$

where $\overline{w}$ is the prescribed unperturbed vertical velocity that the source term tries to attain, $w(\boldsymbol{x})$ is the vertical velocity at a given point $\boldsymbol{x}$ and $\nu_r(z)$ is an activation function, defined as

$$\nu_r(z) = \alpha_r \left[1 - \cos\left(\frac{\pi}{2}\frac{z - z_s}{z_e - z_s}\right)\right] \tag{2}$$

with $z_s$ and $z_e$ the start and end heights of the Rayleigh damping region, and $\alpha_r$ the proportionality coefficient to be chosen depending on the specific problem. Note that only the vertical velocity is damped, as this is the only source of reflection for the upper boundary. In particular, $w(\boldsymbol{x})$ should be practically zero at the boundary, hence $\overline{w} = 0$. Notably, Lanzilao and Meyers

(2023) also apply the damping to the horizontal velocity components. In the present study such operation is not performed, as horizontal fluctuations are not reflected. This also limits the possible counteraction given by the source terms in those cells where the Rayleigh damping overlaps with the fringe region. However, we follow their approach in prescribing $\alpha_r$, which is set to three times the Brunt-Väisälä frequency.

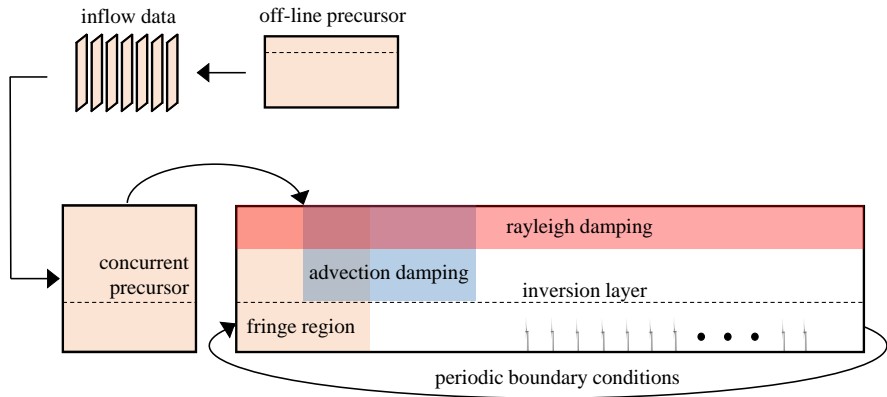

**Figure 1.** Methodological sketch of the AGW-resolved method employing streamwise periodic boundary conditions in the successor domain and a fringe region located at the inlet. The relative location of the Rayleigh damping layer and advection damping region are also shown. The figure is not to scale.

Special care should be paid to the lateral boundaries as well. In the spanwise direction, periodic boundary conditions are used. This ensures no reflections, but in essence renders the solution periodic, allowing waves leaving the domain from one side to re-enter at the opposite side. This is not an issue as long as the domain width is sufficient to ensure that waves reach the spanwise sides far downstream of the wind farm.

In the streamwise direction, the use of periodic boundary conditions implies that the wind farm wake is re-advected at

the inlet. Moreover, Smith (1980) showed that the propagation of wave energy is aligned to the wind direction close to the wave source, i.e. the wind farm. This means that, for conditions practical to large wind farms, energy is radiated almost perpendicularly to the inlet and outlet boundaries, making it impossible to avoid reflections without using damping regions or by massively increasing the domain length. Among these two solutions, the former is usually preferred as it drastically reduces the cell count. For instance, the domain length that allowed to avoid AGW reflections in Lanzilao and Meyers (2023) was of

40 km with a fringe region and 200 km without. In the present study, an inlet fringe region is applied. This stems from the method used in pseudo-spectral codes to enforce an arbitrary inlet boundary condition while still using periodic boundaries in the spectral directions (a requirement imposed by the Fourier transform). The fringe region method is essentially a Rayleigh damping layer where the unperturbed field is heterogeneous in both space and time. While employing periodic boundaries, the flow is slowly brought to an unperturbed state as it transits through the fringe region. This is achieved by applying a source

term on all components of the momentum equation, calculated as

$$s_i^f(\boldsymbol{x}) = \nu_f(x)\left[\overline{u}_i(\boldsymbol{x},t) - u_i(\boldsymbol{x},t)\right] \tag{3}$$

where $u_i(\boldsymbol{x},t)$ are the velocity components at every cell and $\overline{u}_i(\boldsymbol{x},t)$ are the temporally- and spatially-resolved unperturbed flow components, whose calculation will be explained later. The activation function $\nu_f(x)$ only depends on the streamwise coordinate and, following Inoue et al. (2014), it is given by

$$\nu_f(x) = \alpha_f\left[F\left(\frac{x - x_s^f}{\Delta_s^f}\right) - F\left(\frac{x - x_e^f}{\Delta_e^f} + 1\right)\right], \tag{4}$$

with

$$F(x) = \begin{cases} 0, & \text{if } x \leq 0 \\ \left[1 + \exp\left(\frac{1}{x-1} + \frac{1}{x}\right)\right]^{-1}, & \text{if } 0 < x < 1 \\ 1, & \text{if } x \geq 1. \end{cases} \tag{5}$$

The parameters $x_s^f$ and $x_e^f$ are the start and end of the fringe region, respectively, while $\Delta_s^f$ and $\Delta_e^f$ are the distances required to transition from zero to a damping equal to $\alpha_f$, and from $\alpha_f$ back to zero at the fringe start and exit, respectively. The parameter $\alpha_f$ is the fringe coefficient, which has to be tuned depending on the specific case. For instance, potential temperature should also be damped according to Equation (3), where $u_i(\boldsymbol{x},t)$ and $\overline{u}_i(\boldsymbol{x},t)$ are replaced with $\theta(\boldsymbol{x},t)$ and $\overline{\theta}(\boldsymbol{x},t)$, respectively.

The unperturbed state required to compute the momentum and potential temperature source terms in the fringe region is evaluated by conducting a second simulation, referred to as the concurrent precursor, without wind turbines, in a domain coincident to or larger than the fringe region. This has to advance simultaneously with the wind farm simulation, so that the unperturbed state is available at each iteration. The need to solve for a concurrent precursor and the higher cell count due to the inclusion of damping regions represent the major increase in computational cost for the AGW-resolved method. Notably, streamwise periodic boundary conditions in the wind turbine domain allow to use a single fringe region located at the inlet (Stipa et al., 2024a, present) or at the outlet (Lanzilao and Meyers, 2024).

An additional source of contamination of the physical solution is represented by spurious gravity waves that are generated by the fringe region as it tries to force the wave perturbations to zero. This issue has been addressed by Lanzilao and Meyers (2023), who developed the so-called advection damping region, where horizontal advection of vertical velocity is brought to zero to prevent these spurious oscillations to be advected downstream. Specifically, the term $\partial(uw)/\partial x$ is multiplied by

$$\nu_a(x,z) = 1 - \left[F\left(\frac{x - x_s^a}{\Delta_s^a}\right) - F\left(\frac{x - x_e^a}{\Delta_e^a} + 1\right)\right]\mathcal{H}(z - H), \tag{6}$$

where $x_s^a$ and $x_e^a$ are the start and end of the fringe region, respectively, while $\Delta_s^a$ and $\Delta_e^a$ are the distances required to transition from unity to null magnitude of the advection term and from null back to unity at the region start and exit, respectively;

$\mathcal{H}(z-H)$ is the Heaviside function which ensures that this operation is only performed above the boundary layer, in order to leave turbulence unaffected.

In the present paper, the hybrid off-line/concurrent method described in Stipa et al. (2024a) is used. This essentially reduces the computational cost associated with turbulence initialization in the concurrent precursor domain. In fact, while the size of the concurrent precursor has to be equal to or larger than the fringe region, the latter is required to compute the source terms only when the wind turbine simulation is started. Hence, turbulence spin up can be achieved by first running a separate precursor in a reduced domain, referred to as the off-line precursor. In particular, since no gravity waves are expected during this phase, the vertical domain size is such that only a small portion of the free atmosphere is resolved. Moreover, if the ratio between the spanwise size of the concurrent and offline precursor is an integer, off-line precursor data can be prescribed at the inlet of the concurrent precursor by tiling them in the spanwise direction and extrapolating in the vertical. The concurrent precursor is then evolved using inlet-outlet boundary conditions for one flow turnover time, i.e. until it is filled with such pre-calculated turbulent flow. Boundary conditions in the concurrent precursor are then switched to periodic and the simulation becomes self-sustained. The hybrid off-line/concurrent precursor method allows to reach a fully developed ABL within a domain that is sufficient to decorrelate turbulent fluctuations but whose size is not dictated by the wind farm and AGW scales, thus allowing to save computational resources (see Stipa et al., 2024a for more details). In Figure 1, a methodological sketch of the AGW-resolved approach employing the concurrent precursor method is displayed.

### 2.2 AGW-Modeled Approach

As pointed out by Allaerts and Meyers (2017, 2018, 2019) and Lanzilao and Meyers (2022), AGWs in the free atmosphere induce large-scale pressure gradients inside the ABL. The MSC model developed by Stipa et al. (2024b) is based on the concept that the effect of AGW on the wind farm is given by the change in mean velocity produced by this horizontally heterogeneous pressure field, here referred as $p^\star$. Unfortunately, this idea cannot be directly applied to wind farm LES by prescribing $p^\star$ as a separate source term. In fact, for reasons that will be clarified later, the presence of the upper boundary automatically prescribes a certain horizontal pressure gradient inside the ABL such that mass and momentum conservation are satisfied. Specifically, we show below that the vertical streamline displacement $\eta$ prescribed by the presence of the top boundary and the large-scale pressure field $p^\star$ cannot be imposed simultaneously, but are rather interdependent.

The relationship between these two variables can be explained by constructing a simple model based on a perturbation analysis applied to the depth-averaged linearized Navier-Stokes equations. Although this leads to a consistent simplification of the equations proposed by Allaerts and Meyers (2019) and cannot provide an accurate description of the boundary layer flow, this simple model still provides a level of physical insight that is sufficient to elucidate the relationship between the boundary layer displacement $\eta$ and the pressure $p^\star$. First, an infinitely wide wind farm is assumed in the spanwise direction, so that quantities can only change along the streamwise direction (i.e. $\partial/\partial y = 0$). Furthermore, the background flow is assumed to have a null mean spanwise component (i.e. Coriolis force is neglected). The structure of the potential temperature profile is that of a CNBL characterized by lapse rate $\gamma$, inversion strength $\Delta\theta$ and inversion height $H$. The bulk velocity within the boundary layer and the geostrophic wind are referred to as $U$ and $G$, respectively. Similarly to Allaerts and Meyers (2019), the region

below $H$ is divided into two layers, namely the wind farm layer, characterized by a height $H_1$, and the upper layer, of depth $H - H_1$. The depth of the wind farm layer is chosen as twice the hub height, i.e. $H_1 = 2h_{\text{hub}}$. Finally, it is assumed that wind farm and upper layer are characterized by the same background velocity $U$, but at the same time admit different perturbation velocities $u_1$ and $u_2$. With the above simplifications, the 3LM equations derived by Allaerts and Meyers (2019) become

$$
\begin{cases}
U\dfrac{\partial u_1}{\partial x} + \dfrac{1}{\rho}\dfrac{\partial p^\star}{\partial x} = -\dfrac{C}{H_1}u_1 - \dfrac{f_x}{H_1} \\[2mm]
U\dfrac{\partial \eta_1}{\partial x} + H_1\dfrac{\partial u_1}{\partial x} = 0,
\end{cases}
\tag{7}
$$

for the wind farm layer, and

$$
\begin{cases}
U\dfrac{\partial u_2}{\partial x} + \dfrac{1}{\rho}\dfrac{\partial p^\star}{\partial x} = 0 \\[2mm]
U\dfrac{\partial \eta_2}{\partial x} + H_2\dfrac{\partial u_2}{\partial x} = 0,
\end{cases}
\tag{8}
$$

for the upper layer, where $C = 2u^{*2}/U$. It should be noted that $\eta_1 + \eta_2 = \eta$, i.e. the total vertical displacement of the pliant surface initially located at $H$. This, at steady state, coincides with the flow streamline through $H$ far upstream, and can be thought as both the inversion layer or ABL vertical displacement.

Rewriting the system in terms of $\eta$ reads

$$
\begin{cases}
U\dfrac{\partial u_1}{\partial x} + \dfrac{1}{\rho}\dfrac{\partial p^\star}{\partial x} = -\dfrac{C}{H_1}u_1 - \dfrac{f_x}{H_1} \\[2mm]
U\dfrac{\partial u_2}{\partial x} + \dfrac{1}{\rho}\dfrac{\partial p^\star}{\partial x} = 0 \\[2mm]
U\dfrac{\partial \eta}{\partial x} + H_1\dfrac{\partial u_1}{\partial x} + H_2\dfrac{\partial u_2}{\partial x} = 0.
\end{cases}
\tag{9}
$$

To complete the system, an extra equation is added that relates the vertical inversion displacement to the pressure anomaly felt inside the boundary layer due to the increase or decrease in weight of the air column overtopping a given $x$ location. This can be expressed in Fourier space by means of linear theory (Nappo, 2012; Lin, 2007) as

$$
\frac{1}{\rho}\hat{p}^\star = \Phi\hat{\eta},
\tag{10}
$$

where the hat denotes Fourier coefficients and $\Phi$ is the so-called complex stratification coefficient, which accounts for pressure anomalies generated by both the inversion layer displacement (surface waves) and the resulting perturbations aloft (internal waves). We refer to Smith (2010) for the definition of such function; in this context it is sufficient to notice that all the physics related to AGWs and thermal stratification enters the system through the complex stratification coefficient $\Phi$, while Equation (9) does not contain any stability-related term.

Equation (9) and Equation (10) form a fully determined system, which can be easily solved upon transforming Equation (9) into Fourier space. In particular, Equation (9) describes the flow physics below $H$, while Equation (10) refers to the flow in the free atmosphere. It can be observed that the pressure field $p^\star$, which satisfies both Equations (9) and (10), is the one reconciling

momentum and mass conservation inside the boundary layer with pressure anomalies due to overtopping density differences produced by a determined vertical displacement of the pliant surface at $H$. Specifically, $\eta$ represents the coupling variable between the ABL and the free atmosphere, i.e. the neutral and stratified regions of the flow, respectively, under CNBL. Now, focusing only on the flow below $H$, i.e. on Equation (9), it is evident how the pressure gradient induced by AGWs — and its effects on the velocity — could be readily obtained without any knowledge about free atmosphere stratification if the correct inversion displacement $\eta$ was somehow known *a priori*. The same reasoning can be applied to the full 3LM equations derived by Allaerts and Meyers (2019) in the first two layers by simply noticing that the number of unknowns is reduced by one. Extending this idea to LES, the knowledge of $\eta$ can be used to vertically deform the top boundary of the computational domain when this is initially located at $H$. Since a slip condition is usually applied here, deforming the upper boundary alters the mean flow streamlines in a manner that is consistent with the inversion-layer displacement, allowing the mean AGW induced pressure gradient to be automatically recovered within the ABL. In summary, AGWs effects produced by different stability conditions can be easily modeled by using their corresponding $\eta$ field to vertically deform the upper boundary. By doing so, the LES can be conducted only in the turbulent part of the flow capped below the start of the stable flow region, where the wind farm is located.

In the present study, the top boundary is placed at the inversion center and the MSC model is used to compute $\eta$. The vertical displacement is linearly distributed to the underlying mesh points, deforming the computational grid before starting the simulation. This means that, at each horizontal location, the mesh point located at the wall is not displaced while the furthest is moved vertically by $\eta$. The grid points in between will be displaced from zero to $\eta$, depending on their distance from the wall, thus causing vertical cell stretching.

The case where the top boundary is a flat surface corresponds to the rigid lid limiting solution. In particular, while this differs from the actual solution with atmospheric gravity waves, it still models — to a certain extent — both global blockage and flow acceleration within the wind farm produced by flow confinement inside the boundary layer. We highlight that flow confinement and free atmosphere stability effects are different ways to refer to the same physical manifestation. In fact, in light of the unique relationship between pressure and ABL displacement, stability effects determine an inversion displacement such that pressure perturbations induced by flow confinement and by AGW are equivalent, i.e. they satisfy both Equations (9) and (10). For this reason, the rigid lid assumption models global blockage to a certain degree, as the flow is indeed confined. However, the mechanism under which such confinement happens disregards gravity waves by neglecting the inversion perturbation field that complies with the actual potential temperature structure.

As a further consideration on the AGW-modeled technique, since the overall pressure disturbance is fully determined by the inversion displacement, any spatially-varying source term imposed in the form of a pressure gradient will not produce any effect on the simulation results, but rather change the significance of the pressure variable such that the overall pressure disturbance complying with the imposed streamline displacement is retained.

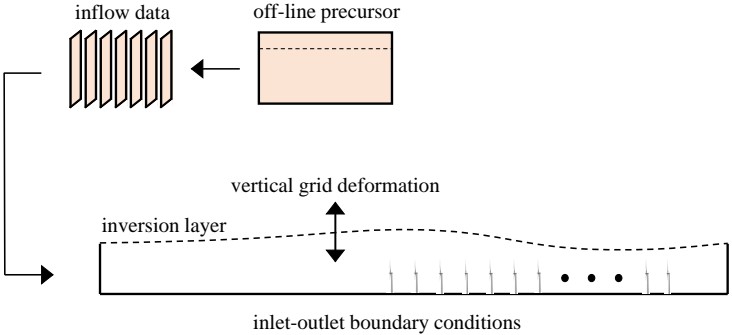

**Figure 2.** Methodological sketch of the AGW-modeled method. The figure is not to scale.

The developed approach, sketched in Figure 2, is convenient for at least three reasons. First, it substantially reduces the computational cost by eliminating the need for damping regions and the requirement of a domain that is large enough to vertically resolve AGWs. Secondly, it does not require to run a concurrent precursor simultaneously to the wind farm simulation. Third, under CNBLs it eliminates the need to solve for a potential temperature transport equation as the flow below $H$ is neutral. This condition is only violated very close to the top boundary, where discrepancies in turbulent fluctuations produced by the absence of stability and by the physical boundary are deemed acceptable as they happen away from the wind farm. Moreover, at the inversion height fluctuations are naturally close to zero, as this roughly coincides with the top of the boundary layer.

A limitation of the proposed method is that the accuracy of the large-scale pressure gradient produced by displacing the top boundary is dependent on the accuracy of the MSC model in predicting the overall AGW physics. Stipa et al. (2024b) showed that the pressure disturbance produced by the MSC model agrees well with AGW-resolved wind farm LES simulations for different values of capping inversion strength. Another limitation is given by the fact that the MSC model has not been tested for values of $H/H_1$ that are less than or equal to one, a realistic condition for modern large turbines. This corresponds to a situation where the turbine top tip almost pierces the inversion layer, with consequent disappearance of the upper layer. Devesse et al. (2023) developed an alternative strategy to the one used in the MSC model to couple the 3LM of Allaerts and Meyers (2019) and the Bastankhah and Porté-Agel (2014) wake model, which also uses the 3LM to address AGW effects. When validating this new model against wind farm LES characterized by $H = 150, 300, 500$ and $1000$ m and $h_{hub} = 119$ (Lanzilao and Meyers, 2024), the authors excluded those LES cases with $H/H_1 = 0.63$ ($H = 150$ m). Among the remaining cases, the model showed the highest deviation from the LES when $H/H_1 = 1.26$ ($H = 300$ m). As also the MSC model uses the 3LM to model AGW effects, these results suggest that the MSC model will loose accuracy when $H/H_1 \lesssim 1.5$. In the present manuscript, the dependency of the proposed technique to the ratio $H/H_1$ is not investigated and this number is fixed to 2.78. Although this is a limitation of the MSC model, if $\eta$ could be evaluated by different means (e.g. with a coarser AGW-resolved LES employing a simple canopy model) at a height located above the inversion layer, the AGW modeling approach

could be used for small $H/H_1$ ratios by placing the upper boundary a few hundreds meters into the free atmosphere and by including the potential temperature transport equation. A related limitation applies to those cases characterized by an unsteady

flow in the free atmosphere, as the MSC model assumes steady state conditions.

## 3  Suite of Simulations

To verify the validity of the proposed approach, we use the two LES simulations available from Stipa et al. (2024b). These correspond to a subcritical and a supercritical regime of interface waves within the inversion layer and are characterized by damping regions and a domain size that is sufficient to resolve AGWs in the free atmosphere. For this reason, they are

325 referred to as AGW-resolved cases in the present study. Each case is then compared to its AGW-modeled counterpart, where the technique proposed in Section 2.2 is applied. Once validated, the AGW-modeled approach is leveraged to simulate a case corresponding to the rigid lid limiting solution, where the top boundary is not associated with any vertical displacement. This analysis is motivated by the fact that the rigid lid enforces dependency of the inversion layer height while discarding the full AGW solution, making it an appealing approximation in the context of reduced-order engineering parametrizations. As it will

be shown, its estimates on blockage are in some cases better than those of a conventional wake model combined with a local induction model, which only account for local turbine induction.

### 3.1  AGW-Resolved Simulations

The subcritical and supercritical regimes of the AGW-resolved CNBL simulations are obtained by setting the inversion strength to 7.312 K and 4.895 K, respectively. Table 1 summarizes the remaining input parameters, namely the reference velocity $u_{\text{ref}}$ at

335 the reference height $h_{\text{ref}}$ (chosen as the hub height), the reference potential temperature $\theta_0$, the lapse rate $\gamma$, the inversion height $H$ and the equivalent roughness length $z_0$. The Coriolis parameter $f_c$ corresponds to a latitude of 41.33 deg. The simulated wind farm has a rectangular planform, with 20 rows and 5 columns organized in an aligned layout. The first row is located at $x = 0$, and extends from 300 m to 2700 m in the spanwise direction. This determines a lateral spacing of 600 m (4.76 D), while streamwise spacing is set to 630 m (5 D). Wind turbines correspond to the NREL 5-MW reference turbine, and are equipped

with the angular velocity and pitch controllers described in Jonkman et al. (2009). A very simple yaw controller is also added, which rotates wind turbines independently using a constant rotation speed of $0.5\ {}^{\circ}\text{s}^{-1}$ when flow misalignment exceeds 1 deg. Flow angle for the yaw controller is calculated by filtering the wind velocity at a sampling point located 1 D upstream of the rotor center, using a time constant of 600 s. Turbines are modeled using the actuator disk model (ADM) described in Stipa et al. (2024a), while the tower and nacelle are not included in the simulation. The ADM force projection width is set to 18.75

345 m.

| $u_{\text{ref}}$ [m s$^{-1}$] | $h_{\text{ref}}$ [m] | $\theta_0$ [K] | $\Delta h$ [m] | $\gamma$ [K km$^{-1}$] | $H$ [m] | $f_c$ [s$^{-1}$] | $z_0$ [m] |
|---|---|---|---|---|---|---|---|
| 9.0 | 90 | 300 | 100 | 1 | 500 | $9.6057 \cdot 10^{-5}$ | 0.05 |

**Table 1.** Reference velocity $u_{ref}$ at the reference height $h_{ref}$, reference potential temperature $\theta_0$, inversion width $\Delta h$, lapse rate $\gamma$, inversion center $H$, Coriolis parameter $f_c$ and equivalent roughness height $z_0$ used as input for the finite wind farm simulations presented in this section.

The AGW-resolved simulations employ the hybrid off-line/concurrent precursor method described in Stipa et al. (2024a). For the off-line precursors, the Rampanelli and Zardi (2004) model is used to initialize the potential temperature profile, where $H$ is taken as the center of the capping inversion layer. Off-line precursors for both ABL conditions are advanced in time for $10^5$ s, after which data are averaged for $2 \cdot 10^4$ s. Results from this phase are reported in Appendix A. The off-line precursor domain size is of 6 km $\times$ 3 km $\times$ 1 km in the streamwise, spanwise, and vertical directions respectively. The mesh has a horizontal resolution of 15 m, while in the vertical direction it is graded equally as the concurrent precursor and successor simulations, described later. A driving pressure controller that employs the geostrophic damping method is used to fix the average velocity at $h_{\text{ref}}$ while eliminating inertial oscillations in the free atmosphere. Moreover, a potential temperature controller is used to fix the average potential temperature profile throughout the simulation (both controllers and geostrophic damping use the settings reported in Stipa et al., 2024a). Inflow slices saved during the off-line precursor phase are then used to feed the concurrent precursor for one flow through time (approximately 700 s). Then, boundary conditions in the concurrent precursor domain are switched to periodic and the solution becomes self-sustained. At each successor iteration, velocity and temperature from the concurrent precursor are used to compute the damping terms for the momentum and temperature equations inside the fringe region, located at the inlet of the successor domain. This allows a time-varying turbulent flow to be produced at the fringe exit while eliminating the reintroduction of the wind farm wake at the inlet operated by the periodic boundaries. Moreover, the fringe region allows to damp gravity wave reflections. At the upper boundary, a Rayleigh damping layer is used with a thickness of 12 km, i.e. slightly more than one expected vertical wavelength $\lambda_z$ (this parameter can be estimated as explained in Section 1). Lateral boundaries are periodic, implying that gravity waves induced by the wind farm will interact with their periodic images. This dictates that the domain must be sufficiently large for these interactions to happen far from the wind turbines. The advection damping technique described in Section 2.1 is used to ensure that interactions between fringe-generated and physical gravity waves are not advected downstream but instead remain trapped inside the advection damping region. The Rayleigh damping coefficient $\alpha_r$ is set to 0.05 s$^{-1}$, while the fringe damping coefficient $\alpha_f$ is set to 0.03 s$^{-1}$. The fringe and advection damping functions are given by Equation (4) and Equation (6), respectively, and their parameters are reported in Table 2.

| $x_s^f$ [km] | $x_e^f$ [km] | $\Delta_s^f$ [km] | $\Delta_e^f$ [km] |
|:---:|:---:|:---:|:---:|
| $-20$ | $-15$ | 1 | 1 |

(a) Fringe region parameters.

| $x_s^a$ [km] | $x_e^a$ [km] | $\Delta_s^a$ [km] | $\Delta_e^a$ [km] |
|:---:|:---:|:---:|:---:|
| $-18$ | $-11$ | 1 | 1 |

(b) Advection damping region parameters.

**Table 2.** Fringe and advection damping region parameters.

The domain size of the AGW-resolved successor cases is 40 km $\times$ 21 km $\times$ 28 km in the streamwise, spanwise and vertical direction respectively, discretized with 1554 $\times$ 1194 $\times$ 345 cells. All directions are graded to reach a mesh resolution of 30 m $\times$ 12.5 m $\times$ 10 m around the wind farm, as reported by Stipa et al. (2024b). The concurrent precursor mesh coincides with the portion of the successor domain located inside the fringe region. As a consequence, it extends for 5 km $\times$ 21 km $\times$ 28 km. Here, the mesh resolution is identical to the successor mesh.

## 3.2 AGW-Modeled Simulations

The AGW-modeled simulations feature the same horizontal domain size and discretization as the AGW-resolved cases. Conversely, the vertical domain size is set to 500 m, which is coincident with the unperturbed inversion layer height. Since the fringe region, Rayleigh and advection damping layers are not required, inlet-outlet boundary conditions are used in the streamwise direction. Due to the fact that the same inflow data used in the AGW-resolved cases was not available because it has been generated at run time within the concurrent precursor, two additional off-line precursor simulations have been conducted, corresponding to the subcritical and supercritical conditions defined in Section 3.1. These are first run for $10^5$ s, after which data are averaged for $4 \cdot 10^4$ s and inflow sections are saved at each iteration to be used as inlet boundary conditions in the wind farm successors. These additional precursor cases are characterized by an enlarged spanwise domain size of 21 km – coincident with the successor cases – to avoid the spanwise periodization of the inflow data that characterizes the initial condition for the AGW-resolved cases in Stipa et al. (2024b). As reported by the same authors, this led to turbulent streaks that slowed down the convergence of turbulence statistics. To further address this issue, a spanwise shift velocity of 1 m s$^{-1}$ is applied to the inflow data in the AGW-modeled successor simulations, with the objective of enhancing statistics convergence. Instead of being added to the inflow velocity field, such shift velocity is used to physically move the inflow data along the spanwise direction so that the average wind direction remains unaffected. Results from these off-line precursor characterized by an enlarged domain are reported in Appendix A, together with their comparison with the off-line precursors conducted for the AGW-resolved simulations. The inflow data is then mapped at the successor inlet patch by means of bi-linear interpolation, further interpolating at the desired time value from the two closest available times. AGW-modeled simulations are progressed in time for $4 \cdot 10^4$ s, using the entire inflow database. This differs from the AGW-resolved cases, which have been only progressed for $2 \cdot 10^4$ s. Since the unperturbed flow corresponds to a CNBL, we do not solve for potential temperature, as this is constant throughout the simulation domain. As explained in Section 2.2, stability effects on the ABL flow are embedded in the applied vertical displacement of the inversion layer, which imposes the corresponding pressure perturbation.

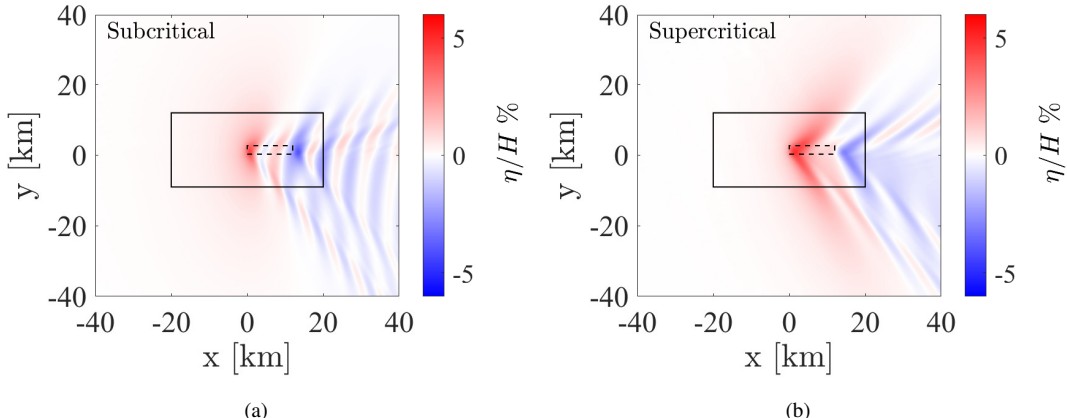

**Figure 3.** Inversion displacement as a percentage of the boundary layer height for (a): subcritical case and (b): supercritical case. The LES domain is identified by the continuous rectangle, while the wind farm is represented by the dashed rectangle.

As previously mentioned, this is calculated using the MSC model. The input parameters for the MSC model are calculated using the fully-developed off-line precursors of the AGW-resolved cases to ensure consistency between these and the AGW-modeled simulations. Although the input parameters required by the MSC model are detailed in Stipa et al. (2024b), they are also reported in Table 3 for completeness. The resulting inversion displacement fields are displayed in Figure 3. For more details about the MSC model setup the reader is referred to Stipa et al. (2024b), where the same atmospheric conditions have been investigated.

Regarding the simulation corresponding to the rigid lid approximation, the velocity inflow data from the subcritical case has been used to prescribe the inlet boundary condition, while the top boundary, also located at $H = 500$ m, has not been displaced.

| input parameter | Subcritical | Supercritical | |
|---|---|---|---|
| $g$ | | 9.81 | [m s$^{-2}$] |
| $\rho$ | | 1.225 | [kg m$^3$] |
| $H, H_1, H_2$ | | 500, 180, 320 | [m] |
| $\Delta\theta$ | 7.312 | 4.895 | [K] |
| $\theta_0$ | | 300 | [K] |
| $\gamma$ | | 1 | [K km$^{-1}$] |
| $\phi$ | | 41.33 | [°] |
| $z_0$ | | 0.05 | [m] |
| $u^*$ | | 0.43 | [m s$^{-1}$] |
| $\nu_{t,1}, \nu_{t,2}$ | | 9.37, 6.19 | [m$^2$s$^{-1}$] |
| $(U_1, U_2, U_3)$ | (8.31, 10.07, 9.77) | (8.41, 10.32, 10.16) | [m s$^{-1}$] |
| $(V_1, V_2, V_3)$ | $(-0.05, -0.78, -4.49)$ | $(0.09, -0.2, -4.41)$ | [m s$^{-1}$] |
| $\left\|\tau|_{z=0}\right\|, \left\|\tau|_{z=H_1}\right\|$ | | 0.19, 0.11 | [m$^2$s$^{-2}$] |
| $TI_\infty$ | | 0.09 | [$-$] |

**Table 3.** MSC model input parameters for the subcritical and supercritical case. The parameters $\nu_{t,1}$ and $\nu_{t,2}$ are the deep-averaged effective viscosities in the wind farm and upper layers, evaluated using the Nieuwstadt (1983) model; $U_i$ and $V_i$ (with $i = 1:3$) are the streamwise and spanwise velocity components, respectively, deep-averaged from the AGW-resolved off-line precursors within layer $i$; $TI_\infty$ is the hub-height freestream turbulence intensity.

## 4 Results

In the following, the accuracy of the proposed AGW-modeled method is first verified against AGW-resolved simulations corresponding to Stipa et al. (2024b), for both subcritical and supercritical conditions, in Section 4.1. Then, in Section 4.2, the implications of employing the rigid lid approximation (Smith, 2023) in terms of our ability to capture global blockage effects are investigated. The latter corresponds to an infinitely high free atmosphere stability, obtained by modeling the inversion layer as a rigid lid. As a consequence the resulting horizontal pressure gradient solely responds to the requirement of mass conservation inside the boundary layer.

For the cases presented in this manuscript, the AGW-modeled technique requires a domain with $\approx 12.7\%$ of the grid cells used for the AGW-resolved simulations. Although the domain used for the off-line precursors is larger for the AGW-modeled cases, this is not a requirement of the developed approach. In fact, a smaller domain with lateral inflow periodization technique is probably sufficient if combined with the spanwise shift used in this manuscript to accelerate statistics convergence.

The AGW-resolved and AGW-modeled cases consist of 15 000 s and 35 000 s of available data, respectively, following the establishment of a statistically-steady flow in the successor simulations. However, when comparing turbine quantities, time

averaging has been performed for 15 000 s in both cases. Specifically, since the AGW-modeled cases used different precursor time histories from the AGW-resolved counterparts, the start of the time averaging window has been shifted in time until the row-averaged freestream velocity at the first turbine row matched those from the corresponding AGW-resolved simulations. This allows to compare the two approaches eliminating any bias in freestream wind speed produced by the different large-scale turbulent structures in the two cases. More details on this procedure and on its motivation are reported in Appendix B. This approach is only followed when comparing turbine power and thrust between the AGW-resolved and AGW-modeled simulations. Elsewhere, results are always averaged on the entirety of the available time history.

## 4.1  Model Verification

While local blockage is given by the combination of individual turbine induction effects, global blockage can be explained as the flow responding to the pressure gradient induced by the mean ABL displacement (Stipa et al., 2024b). In turn, this results from the vertical velocity perturbation operated by the wind farm and the corrective response provided by buoyancy forces. The induced perturbation pressure field is heterogeneous in space and features an unfavorable pressure gradient region upstream of the farm and a favorable region that extends throughout most of the cluster. Downstream and around the wind farm, the perturbation pressure field is strongly dependent on the strength of the inversion, the free atmosphere lapse rate and the geostrophic wind.

In light of the critical role played by the perturbation pressure field, we first compare the mean pressure variations resulting from the AGW-modeled approach with those obtained by resolving gravity waves in the free atmosphere. Figure 4 plots the streamwise distribution of pressure perturbation, averaged over the wind farm width and in the upper layer (from $H_1$ to $H$), for both the subcritical and supercritical cases, using the AGW-resolved and AGW-modeled approaches. For completeness, the pressure variation resulting from the MSC model is also reported. In both atmospheric states, all models predict similar trends in the pressure perturbation distribution. As explained in Section 2.2, the latter is a function of the imposed vertical boundary layer displacement for the AGW-modeled simulations, while it naturally arises from the free atmosphere solution in both the AGW-resolved approach and the MSC model. Subcritical conditions produce larger pressure gradients if compared to the supercritical ABL state, both unfavorable upstream and favorable inside the wind farm. Moreover, lee waves in subcritical conditions (visible in Figure 3), induce pressure oscillations on a wavelength that is lower than the wind farm length. As can be appreciated from Figure 4, these oscillations are superimposed on the favorable pressure gradient inside the wind farm and lead to oscillations in the background velocity field, power variations throughout the wind farm and an intermittent wake recovery downstream.

Differences in the pressure disturbance predicted by the AGW-resolved, AGW-modeled and MSC results can be explained as follows. First, with reference to the AGW-modeled and MSC model results, differences in the pressure disturbance are attributable to the simplifications made in the MSC model such as linearity, the simpler parametrization of the wind farm and of the turbulent momentum fluxes, as well as the lack of resolved turbulence. As a consequence, even though $\eta$ is identical between the two approaches, these aspects inevitably affect the momentum budget, leading to differences in both velocity and pressure. Regarding the AGW-resolved and AGW-modeled approaches, differences in the pressure field arise from differences

between the imposed $\eta$ distribution (AGW-modeled) and the $\eta$ distribution that develops naturally (AGW-resolved), although the two methods share the same accuracy inside the ABL. Notably, both the MSC and AGW-modeled approaches are able to capture the different trends in pressure disturbance arising from the potential temperature structure in the subcritical and supercritical cases. This is achieved at a drastically lower computational cost than the AGW-resolved method, in our opinion justifying the pressure deviations observed in Figure 4, bounded below a maximum value of $1.5\rho u_{\text{ref}}^2$. Moreover, the arguments provided in Section 2.2 regarding the role of the pressure variable are confirmed, as its dependence on free atmosphere stability is captured in the AGW-modeled approach without solving for the potential temperature equation. Finally, the mismatch in $p'$ between the AGW-modeled and AGW-resolved results near the end of the domain in Figure 4 is produced by the fringe region employed at the domain inlet in the AGW-resolved simulations. As explained in Section 2.2, the fringe region removes the wind farm wake by forcing the flow to adhere to the concurrent-precursor solution at the fringe exit. In doing so, it modifies the momentum balance, altering the pressure field both inside and immediately upwind of the fringe region, which coincides with the domain exit in Figure 4 owing to the periodic boundary conditions.

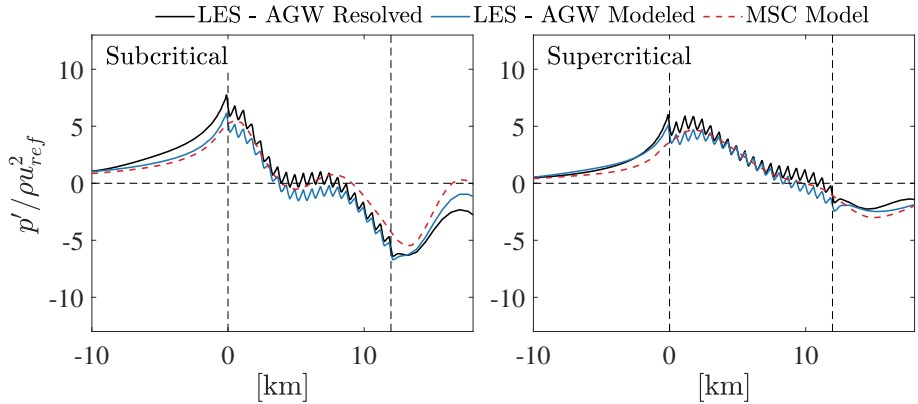

**Figure 4.** Time-averaged streamwise distribution of pressure perturbation, further averaged over the wind farm width (between $y = 0$ and $y = 3000$ m) and in the upper layer (from $H_1$ to $H$), for both the subcritical and supercritical cases. For completeness, the pressure variation resulting from the MSC model is also shown in dashed red. AGW-resolved and AGW-modeled results are shown in black and blue, respectively.

Figure 5 shows the streamwise evolution of hub-height velocity, averaged over time and over the wind farm width, for both the subcritical and supercritical ABL states, for the AGW-resolved and AGW-modeled methods. On the right panel, the metric $(u_{AGWM} - u_{AGWR})/u_{AGWR}$ in percent is also reported, showing the relative percentage error of the velocity predicted by the AGW-modeled with respect to the AGW-resolved approach. Results from the two models are in excellent agreement with each other, indicating that the proposed methodology is able to capture not only blockage, but also the entirety of gravity wave effects on the ABL flow. Velocity reductions extending several kilometers upstream of the wind farm can be observed in both cases, indicating the presence of global blockage. Moreover, the mean velocity deficit in the wind farm wake is captured equivalently between both methods, where the effect of gravity-wave induced pressure gradients on promoting wake recovery

can be observed, especially for the subcritical case. Regarding the error on velocity, the differences between the two approaches are approximately $\pm 5\%$ inside the wind farm and $-1\%$ outside, which is small enough that becomes difficult to state if they are due to the AGW-modeled technique or attributable to the differences in the inflow data. Notably, this seems to be the case

for supercritical conditions in the region upstream of the wind farm, where results from the AGW-resolved method depict a higher velocity than the AGW-modeled simulations.

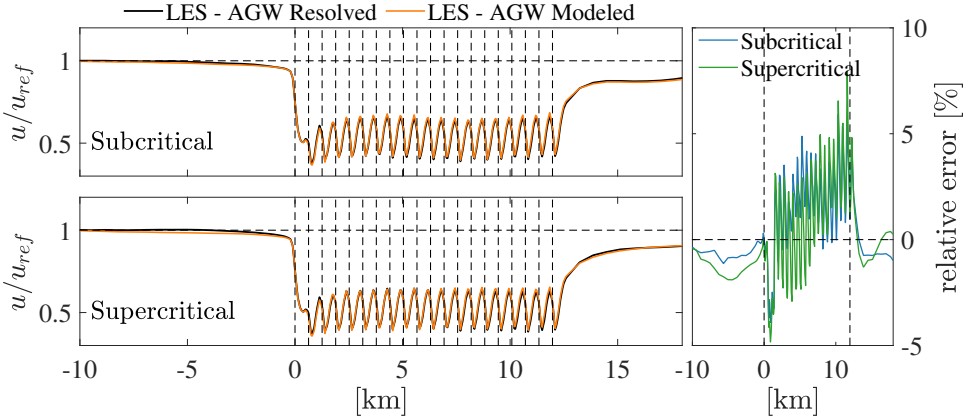

**Figure 5.** Time-averaged hub-height velocity, further averaged over the wind farm width (between $y = 0$ and $y = 3000$ m) for subcritical (top) and supercritical (bottom) conditions for both the AGW-resolved (black) and AGW-modeled (orange) approaches. On the right panel, the relative error of the AGW-modeled with respect to the AGW-resolved method, defined as $(u_{AGWM} - u_{AGWR})/u_{AGWR}$ in percent, is also shown.

In order to verify the accuracy of the proposed method in capturing the turbine thrust and power trends along the wind farm length, the time and row-averaged thrust and power at each wind farm row are plotted in Figure 6 for both the subcritical and supercritical cases. In order to consistently average in time, the approach described in Appendix B has been used. The effect of

480 lee waves aloft in the subcritical case can be appreciated by looking at the large scale oscillations in thrust and power throughout the wind farm. Moreover, the weaker favorable pressure gradient that characterizes supercritical conditions implies lower power towards the wind farm exit. Conversely, the subcritical state is affected by a stronger unfavorable pressure gradient upwind, leading to increased blockage effects. In general, increased blockage also leads to a stronger favorable pressure gradient within the wind farm. However, as demonstrated by Lanzilao and Meyers (2024), whether the net result is beneficial or detrimental

depends on the specific conditions. Overall, it can be stated that the proposed AGW-modeled approach captures the effects of gravity waves on the wind farm power, which is arguably the most important information obtained from a wind farm LES. Table 4 reports the overall wind farm power, as well as the non-local, wake and total wind farm efficiencies $\eta_{nnl}$, $\eta_w$ and $\eta_{tot}$, respectively, as defined by Lanzilao and Meyers (2024), namely

$$\eta_{nnl} = \frac{P_1}{P_\infty} \qquad \eta_w = \frac{P_{tot}}{N_t P_1} \qquad \eta_{tot} = \eta_{nnl}\eta_w, \tag{11}$$

where $P_1$ is the average power at the first wind farm row, $P_{tot}$ is the total wind farm power, $N_t$ is the total number of wind turbines and $P_\infty$ is the power that an isolated wind turbine would experience in the same operating conditions. Notably, $\eta_{nnl}$ quantifies blockage effects, $\eta_w$ provides information on wake effects, while $\eta_{tot}$ refers to the overall wind farm efficiency. In order to compute $\eta_{nnl}$, the power $P_\infty$ produced by an isolated wind turbine subject to the same conditions is required. To get this information, Lanzilao and Meyers (2024) conducted additional isolated turbine LES that employed the same inflow time history used in the wind farm simulations. In our case, the inflow data is not available for the AGW-resolved simulations, as it has been generated at runtime and is not saved to disk. Hence, we use the turbine data reported in Appendix B of Stipa et al. (2024b) to compute $P_\infty$ by interpolating the turbine power curve using the hub-height freestream velocity experienced at the domain inlet of each simulation, averaged over the entirety of the available samples. Since the data from Stipa et al. (2024b) are evaluated from LESs characterized by uniform inflow and absence of turbulence, the values of $\eta_{nnl}$ and $\eta_{tot}$ likely differ from the figures that would be obtained the same inflow data as the AGW-resolved and AGW-modeled simulations. However the differences in each parameter between the two methodologies can still be compared, highlighting the ability of the proposed method to capture blockage and wake effects.

|  | subcritical | | | | supercritical | | | |
| --- | --- | --- | --- | --- | --- | --- | --- | --- |
|  | P [MW] | $\eta_{tot}$ | $\eta_{nnl}$ | $\eta_w$ | P [MW] | $\eta_{tot}$ | $\eta_{nnl}$ | $\eta_w$ |
| AGW-resolved | 135.0 | 0.40 | 0.74 | 0.54 | 133.5 | 0.38 | 0.75 | 0.51 |
| AGW-modeled | 139.5 | 0.42 | 0.78 | 0.54 | 133.3 | 0.39 | 0.79 | 0.49 |

**Table 4.** Overall wind farm power obtained from LES in subcritical and supercritical conditions using the AGW-resolved and AGW-modeled techniques. In addition, the total, non-local and wake wind farm efficiencies $\eta_{tot}$, $\eta_{nnl}$ and $\eta_w$, respectively, are reported. The value of $P_\infty$ required to compute $\eta_{nnl}$ has been obtained from the data reported in Appendix B of Stipa et al. (2024b).

The obtained values of $\eta_w$ agree better than $\eta_{nnl}$ between the AGW-resolved and AGW-modeled cases, for both subcritical and supercritical conditions. The AGW-modeled method captures both the increase in blockage effects from supercritical to subcritical conditions ($\eta_{nnl}$ from 0.79 to 0.78) as well as the efficiency improvement owing to the favorable pressure gradient throughout the wind farm in the subcritical case ($\eta_w$ from 0.49 to 0.54). The fact that values of $\eta_{nnl}$ differ more than $\eta_w$ between the two methodologies suggests that the variations in total wind farm power — and consequently in $\eta_{tot}$ — are mostly due to a power bias at the first row rather than an inconsistency distributed over the entire wind farm. However, we believe that differences in the turbulent inflow data between the AGW-resolved and AGW-modeled simulations are the main cause of such power bias at the first row, considering the extremely good agreement on the hub-height velocity observed in Figure 5.

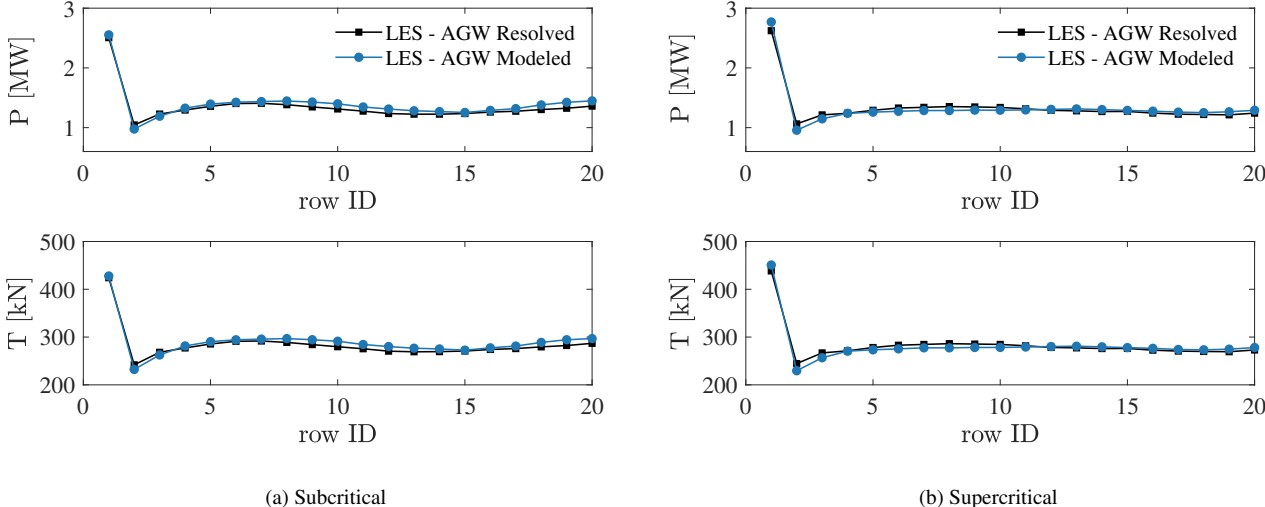

**Figure 6.** Comparison of row-averaged thrust and power distributions for the subcritical (a) and supercritical (b) cases. Time averaging is performed as described in Appendix B. AGW-resolved data are shown in black, while results from the AGW-modeled simulations are depicted in blue.

## 4.2 Implications of the Rigid Lid Approximation

In the present section, the proposed methodology is leveraged to assess the implications of the rigid lid approximation in evaluating global blockage effects. The simpler formulation with respect to the full AGW solution renders the rigid lid assumption attractive for its potential use in future fast engineering models. However, its relation with the full AGW solution has not yet been assessed in detail, together with its differences when compared with a truly neutral case, where stratification is absent.

To enforce the rigid lid approximation within LES, we employ the AGW-modeled technique with no vertical displacement of the top boundary, which is located at $500$ m. Figure 7 shows the mean streamwise distributions of hub-height velocity and depth-averaged pressure between $H_1$ and $H$, further averaged over the width of the wind farm, i.e. from $y = 0$ m to $y = 3000$ m. In particular, it compares the subcritical, supercritical and rigid lid cases, all obtained using the AGW-modeled approach. A close up view of the blockage region within 1 km upstream of the wind farm is also reported. First, by looking at the pressure gradient, it can be noticed how each case is characterized by an anti-symmetric pressure distribution, with maximum and minimum at the wind farm start and exit, respectively. Moreover, the rigid lid approximation is characterized by the lowest values of favorable and unfavorable pressure gradients upstream and inside the wind farm, respectively. As a consequence, while the rigid lid approximation features global blockage, this is less pronounced than both the subcritical or supercritical conditions. Regarding wake recovery, results from the rigid lid case exhibit the highest deficit. Overall, while it is clear that the rigid lid approximation differs from the full gravity-wave solution in terms of pressure perturbations, wake recovery and farm blockage, our results suggest that flow confinement associated with a homogeneous — instead of heterogeneous — inversion height may be responsible for the majority of global blockage effects. This concept will be expanded further in this section.

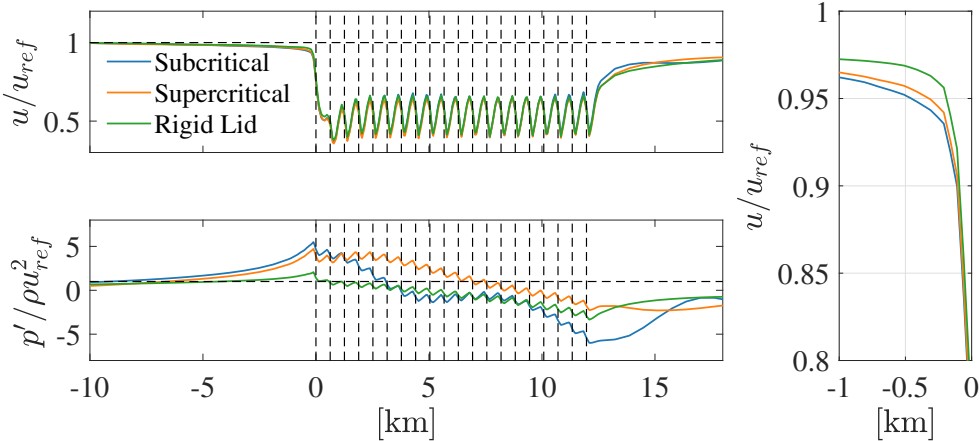

**Figure 7.** Comparison of velocity magnitude (top) and pressure (bottom) between subcritical, supercritical and rigid lid cases. On the right panel, a magnification of the differences in velocity magnitude in the wind farm induction region is reported. Data correspond to the AGW-modeled simulations and are averaged both in time (from $105\,000$ s to $140\,000$ s) and along the wind farm width (from $y = 0$ m to $y = 3000$ m). Pressure data is further averaged vertically between $H_1$ and $H$.

Regarding the mean row-averaged power distributions depicted in Figure 8, it can be stated that, for the simulated conditions, results obtained using the rigid lid approximation are not far from the subcritical and supercritical figures. In fact, referring to the quantitative data reported in Table 5, overall wind farm power from the rigid lid case seems in agreement with the simulations featuring AGW effects. The rigid lid case under-estimates power by $1.5\%$ compared to the subcritical case and overestimates power by $3\%$ with respect to the supercritical case. This trend agrees with the underlying hypotheses of the approximation, where the stronger the potential temperature jump across the inversion layer, the more it behaves as a rigid lid.

|  | P [MW] | rel. difference [%] |
|---|---|---|
| subcritical | 139.5 | -1.5 |
| supercritical | 133.3 | 3.0 |
| rigid lid | 137.4 | |

**Table 5.** Overall wind farm power obtained from LES simulations in subcritical and supercritical conditions, as well as employing the rigid lid approximation. The relative difference of the latter with respect to the first two cases is also reported. Time averaging has been performed in all cases from $105\,000$ s to $140\,000$ s.

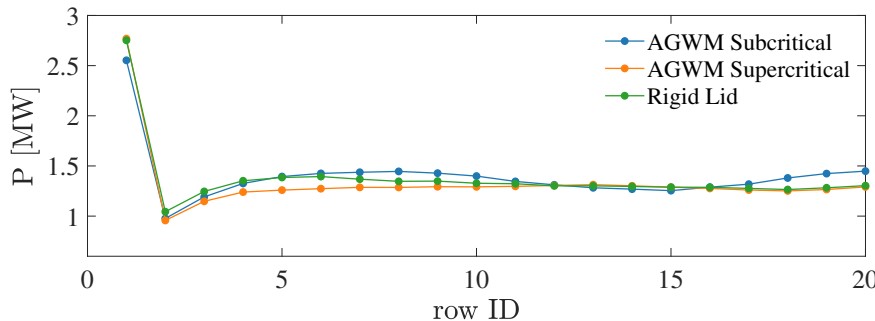

**Figure 8.** Row-averaged power from the subcritical, supercritical and rigid lid cases. Data relative to the subcritical and supercritical conditions correspond to the AGW modeled simulations. Time averaging has been performed in all cases from $105\,000$ s to $140\,000$ s.

However, the LES results do not provide a clear picture on the relation between the full AGW solution and the rigid lid approximation, where the effect of $\Delta\theta$ and $\gamma$ is removed. Notably, removing also the effect of $H$ leads to considering a fully neutral boundary layer, where vertical streamline displacement is not constrained in any way. To further investigate the relation between these three conditions, the MSC model has been used to run a parametric analysis where $\Delta\theta$ and $\gamma$ have been individually varied from 1 to 20 K km$^{-1}$ and 0 to 20 K, respectively. When varying $\Delta\theta$, $\gamma$ has been set to 1 K km$^{-1}$ to match the LESs conducted in this paper. Similarly, when varying $\gamma$, $\Delta\theta$ has been set to 7.312 K and 4.895 K, corresponding to the subcritical and supercritical conditions in this paper, respectively. In addition, two simulations corresponding to the fully neutral and rigid lid cases have been conducted. In the first, $\Delta\theta$ and $\gamma$ have been set to zero, while in the latter they have been set to 1000 K and 1000 K km$^{-1}$, respectively. The rest of the input parameters are identical to those reported in Table 3. Notably, all cases feature a local blockage model as described in Stipa et al. (2024b). For each run, the non-local, wake and total wind farm efficiencies are evaluated. The power of an isolated wind turbine in the same conditions has been calculated by running an additional MSC simulation where the local induction model has been removed from the fully neutral setup. This neglects any kind of blockage effect and all first row turbines produce the same power, taken as $P_\infty$.

For this analysis, it is worthwhile to recall the definitions of the interface Froude number $F_r$ and of the parameter $P_N$, previously defined by Smith (2010), which regulate the physics and magnitude of interface and internal waves, respectively. These can be calculated as

$$F_r = \frac{U_b}{\sqrt{g'H}}, \tag{12}$$

$$P_N = \frac{U_b^2}{NHG}, \tag{13}$$

where $g' = g\Delta\theta/\theta_0$ is the reduced gravity, and $U_b$ is the bulk velocity inside the boundary layer. Subcritical and supercritical conditions are identified by $F_r < 1$ and $F_r > 1$ respectively, while the importance of internal waves reduces as $P_N$ increases.

In Figure 9, the effect of varying $\Delta\theta$ on the wind farm efficiencies is shown. In the top axis, the inversion strength is converted to $F_r$ using Equation (12), where $U_b$ is calculated according to Allaerts and Meyers (2019). For low values of $\Delta\theta$, the non-local efficiency approaches that of a truly neutral case, which is only affected by the combination of individual turbine

induction effects. Conversely, when $\Delta\theta$ is large, $\eta_{nnl}$ approaches the rigid lid solution. The transition from these two cases is strongly non linear, and has a minimum around $F_r = 1$, identified by the vertical continuous line. Interestingly, the minimum of the $\eta_{nnl}$ occurs for a value of $F_r$ slightly lower than unity. For instance, by simplifying the depth-averaged linearized Navier-Stokes equations, Allaerts and Meyers (2019) showed that the second derivative of the total vertical ABL displacement along the streamwise direction $\partial^2\eta/\partial x^2$ is multiplied by a factor $(-1 + F_r^{-2} + P_N^{-1}\mathcal{G})$ (the reader is referred to Allaerts and Meyers, 2019 for the definition of the convolutional operator $\mathcal{G}$), where $F_r$ is defined by Equation (12). On one hand, this shows evidence that using the bulk ABL velocity as the characteristic velocity scale for $F_r$ is based on mathematical grounds. On the other hand, as previously noticed by Smith (2010), the term $(-1 + F_r^{-2} + P_N^{-1}\mathcal{G})$ produces a singularity when $P_N \to \infty$ (no internal waves). Conversely, when internal waves are present, energy is moved away from the source and the singularity disappears. Hence, referring to a condition where both $\Delta\theta$ and $\gamma$ are non-zero, our results seem to suggest that the maximum blockage may be observed at $F_r = 1 - P_N^{-1}\mathcal{G}$ ($\mathcal{G}$ is positive upstream the wind farm) instead of $F_r = 1$, thus exhibiting a dependence on $\gamma$. This behavior can be also noticed from Smith (2010) but has not been mentioned nor discussed further. For our specific case, the minimum of $\eta_{nnl}$ corresponds to $\Delta\theta \approx \theta_0 G^2/(gH)$, i.e. $F_r = 1$ when this is evaluated with $G$ instead of $U_g$. Although this may only hold for the conditions adopted in Figure 9, the drift in the minimum of $\eta_{nnl}$ to lower values of $F_r$ as $\gamma$ increases seems to be a general conclusion, as shown later in Figure 11. Still referring to Figure 9, the truly neutral case is characterized by the highest overall non-local efficiency, while the rigid lid approximation seems to represents a limiting solution for $\Delta\theta \to \infty$. In fact, both higher and lower values of $\eta_{nnl}$ are obtained for different values of $\Delta\theta$ when considering the full AGW solution. The wake efficiency depicts a reversed behavior, with fully neutral conditions characterized by the lowest overall $\eta_w$. Regarding the total wind farm efficiency, this decreases from a value of $\approx 0.44$ in the truly neutral case to $\approx 0.43$ around $F_r = 1$. For increasing values of $\Delta\theta$, $\eta_{tot}$ increases again, overshooting the value corresponding to the rigid lid case by a small amount ($\Delta\eta_{tot} \approx 0.0013$) and slowly approaching it from above when $\Delta\theta \to \infty$. However, it is interesting to note that while fully neutral conditions represent the best and worst case scenario for $\eta_{nnl}$ and $\eta_w$, respectively, the rigid lid approximation yields a value of $\eta_{tot}$ that is close to the maximum achievable by the wind farm, making it far less conservative. Interestingly, a similar sensitivity analysis performed by Allaerts and Meyers (2019) did not capture such behavior. In fact, while their analysis considered upstream blockage effects, it did not include the beneficial pressure gradient inside the wind farm. As a result, wind farm power is erroneously correlated with the velocity deceleration experienced upstream (i.e. $\eta_{tot} \propto \eta_{nnl}$).

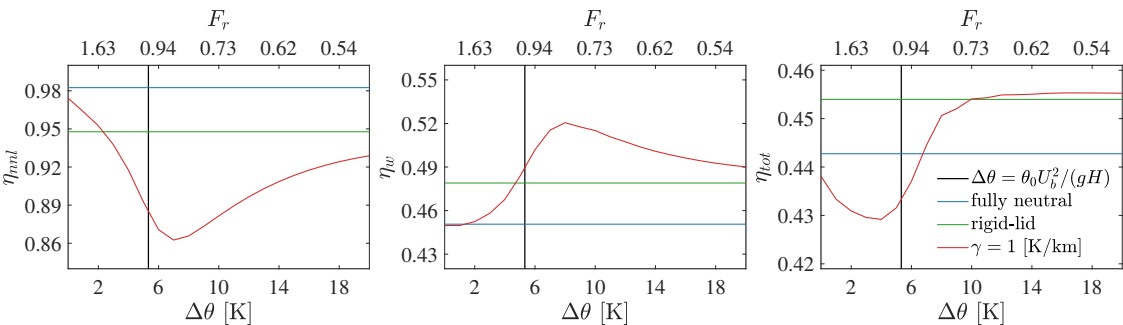

**Figure 9.** Non-local (left), wake (center) and total (right) wind farm efficiency as a function of $\Delta\theta$, when $\gamma = 1$ K km$^{-1}$ (red line). Blue and green lines refer to the fully neutral and rigid lid cases, respectively. Continuous vertical black line refers to $F_r = 1$. The value of $F_r$ as defined by Equation (12) is reported on the top axis.

The behavior of $\eta_{nnl}$, $\eta_w$ and $\eta_{tot}$ when varying $\gamma$ with $\Delta\theta$ fixed is somewhat simpler. This is depicted in Figure 10, where the corresponding value of $P_N$ is also shown on the top axis by converting each $\gamma$ using Equation (13). First, it can be noticed that efficiencies are less sensitive to $\gamma$ than $\Delta\theta$ and their behavior is simpler than that observed in Figure 9. Interestingly, the subcritical case shows little dependency of $\eta_{tot}$ on $\gamma$. In supercritical conditions, the wind farm produces less power than truly neutral conditions for low values of $\gamma$ (also confirmed by our LES simulations), while the efficiency is superior when $\gamma \gtrsim 10$ K km$^{-1}$.

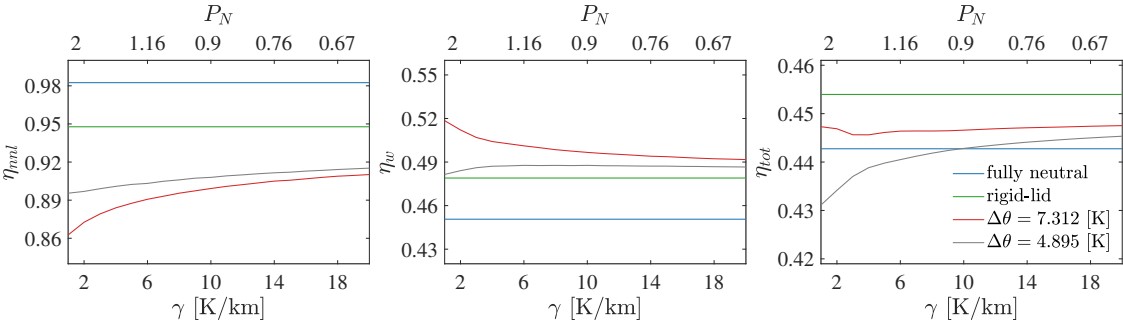

**Figure 10.** Non-local (left), wake (center) and total (right) wind farm efficiency as a function of $\gamma$, with $\Delta\theta = 7.312$ K (red line) and $\Delta\theta = 4.895$ K (gray line). Blue and green lines refer to the fully neutral and rigid lid cases, respectively. The value of $P_N$ as defined by Equation (13) is reported on the top axis.

The parametric study has been then expanded by systematically computing the wind farm efficiencies for all combinations of $\Delta\theta$ and $\gamma$ between 0 and 10 K and 1 and 10 K km$^{-1}$, respectively, with unitary step. The result of this analysis, reported in Figure 11, show that $\eta_{tot}$ increases when both $\Delta\theta$ and $\gamma$ increase, with higher sensitivity to $\Delta\theta$. Notably, conditions where the wind farm extracts more power are also characterized by a large amount of blockage, as testified by the contours of $\eta_{nnl}$. In fact, the decrease in non-local efficiency is compensated by the effect of the favorable pressure gradient inside the wind

farm, which increases $\eta_w$. This conclusion also emerges from the analysis presented by Lanzilao and Meyers (2024), where this effect is shown to be more pronounced as the boundary layer height increases. The location in terms of values of $\Delta\theta$ and $\gamma$ where the minimum $\eta_{nnl}$ is experienced is shown in the $\eta_{nnl}$ contour. This supports the earlier observation that the minimum of $\eta_{nnl}$ corresponds to a $F_r$ slightly lower than one, further decreasing for increasing lapse rate. For instance, these results have been obtained for an aligned wind farm layout with a fixed number of turbine rows and columns, but they are likely to

also depend on the wind farm geometry. However, this is outside of the scope of this paper and represents a subject for future investigation.

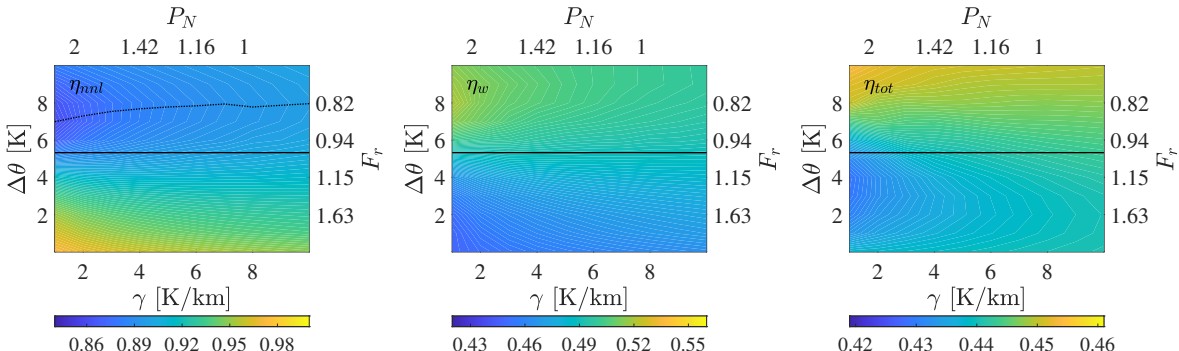

**Figure 11.** Contours of non-local (left), wake (center) and total (right) wind farm efficiency as a function of $\Delta\theta$ and $\gamma$. Continuous horizontal black line refers to $F_r = 1$. The value of $F_r$ as defined by Equation (12) is reported on the right axis, while the value of $P_N$ as defined by Equation (13) is reported on the top axis. The dashed line on the plot of $\eta_{nnl}$ indicates the locus of minima for the non-local efficiency.

To summarize, our results show that the rigid lid approximation yields a total wind farm efficiency that is close to the maximum achievable by the wind farm, while results obtained under a fully neutral ABL are in the middle of the analyzed conditions. This highlights that AGWs play a crucial role in determining the actual value of $\eta_{tot}$, which is in general lower

than that observed when only the effect of $H$ is considered. As a consequence, models employing the rigid lid approximation likely overestimate wind farm power, while those adopting a fully neutral ABL may over- or underestimate wind farm power, depending on the structure of the potential temperature profile.

Finally, the present analysis did not investigate the sensitivity of our results to different values of the inversion height. Nevertheless, based on previous evidence (Lanzilao and Meyers, 2024), AGW effects are expected to fade away with increasing

values of $H$, with both the rigid lid and the full AGW solutions likely approaching the fully neutral case.

## 5   Conclusions

In this study, we introduced an approach that allows the effects of wind farm self-induced atmospheric gravity waves to be modeled without actually resolving these waves in the simulation. The proposed method couples the LES solution below the inversion layer with the MSC model developed by Stipa et al., 2024b. Specifically, the vertical perturbation to the inversion

layer produced by the wind turbines, evaluated with the MSC model, is used to vertically deform the top boundary in the LES domain. Since prescribing the inversion displacement automatically establishes the pressure field below, the resulting LES velocity field contains the influence of gravity waves. If CNBLs are simulated, temperature transport becomes irrelevant as the flow is neutrally stratified inside the domain. AGW feedback with the wind farm is provided by running the MSC model with multiple coupling iterations. The proposed method implies a computational domain that only requires $\approx 12.7\%$ of the cells used in the conventional AGW-resolved method. Moreover, referring to finite volume codes, the simultaneous solution of a concurrent-precursor and the use of a fringe region are not required, as there are no gravity waves in the domain. Thus, incoming ABL turbulence can be prescribed using simple inflow-outflow boundary conditions. More generally, tedious and complex measures to avoid spurious gravity waves reflections, such as the Rayleigh damping layer and the advection damping region, are no longer required.

The proposed approach has been verified against the LES simulations conducted in Stipa et al., 2024b. These are characterized by a setup that allows to resolve AGWs, and correspond to subcritical and supercritical regimes of interfacial waves within the inversion layer. The results show that the proposed method is able to capture the impact of gravity waves on pressure and velocity with good accuracy, correctly estimating blockage effects. Moreover, the row-averaged thrust and power distributions are in good agreement with that of the AGW-resolved approach.

Overall, our analysis shows that the proposed AGW-modeled method allows to model the impact of atmospheric gravity waves on wind farm performance at a reduced computational cost and with sufficient accuracy. A drawback of the approach is that its performance depends on how accurately the MSC model captures the displacement of the inversion layer. Moreover, the MSC model is currently limited to stationary and conventionally neutral boundary layers. For this reason, future work aims at including internal stability and time-dependency into the MSC model, enabling the AGW-modeled method to simulate evolving and arbitrary ABL inflow conditions within LES at a low computational cost. We also plan on extending the verification of the AGW-modeled approach on different atmospheric conditions using data from Lanzilao and Meyers (2024).

Furthermore, the AGW-modeled method has been used to study the implications of adopting the rigid lid approximation (Smith, 2023). The latter neglects the inversion layer displacement produced by gravity waves, thus only considering approximated flow confinement effects. While details due to wind farm gravity waves are expectedly absent, the rigid lid still yields global wind farm blockage, and leads to an overestimation (3% difference) and an underestimation (−1.5% difference) in overall wind farm power when compared to the supercritical and subcritical cases, respectively, employing the AGW-modeled approach. To further investigate the relation between the full AGW solution, the rigid lid approximation, and fully neutral conditions (i.e. absence of stratification), the MSC model has been used to systematically map the error in global wind farm power produced by the rigid lid approximation with different values of inversion strength and free atmosphere stratification. The rigid lid approximation performs worse for supercritical interface wave regimes ($F_r < 1$) and low values of the lapse rate $\gamma$. Conversely, the error reduces with increasing free atmosphere stability, with greater sensitivity to $\Delta\theta$ rather than $\gamma$. Truly neutral conditions yield the lowest blockage and the greatest wake effects. The overall wind farm efficiency given by considering the full AGW solution can be lower or greater than the fully neutral case, depending on the vertical potential temperature profile structure, whereas the rigid lid approximation seems to yield an upper limit for the wind farm efficiency, which increases with

increasing free atmosphere stability. This highlights the importance of considering the potential temperature profile structure when assessing wind farm performance.

## Appendix A: Precursor Simulations

This sections presents the results of the off-line precursor simulations used to generate the turbulent inflow for the AGW-resolved and AGW-modeled simulations. While they share the same input parameters, reported in Section 3, these precursor simulations employ a different domain size in the spanwise direction. Specifically, a domain of $6 \times 3 \times 1$ km has been prescribed in the off-line precursors used to initialize the flow in the concurrent precursor method within the AGW-resolved simulations. Conversely, the off-line precursor conducted to generate the inflow data for the AGW-modeled simulations employed a domain size of $6 \times 21 \times 1$ km. Figure A1 reports the horizontally-averaged vertical profiles of wind speed magnitude, wind angle, potential temperature and shear stress for the subcritical and supercritical cases, for the two different domain sizes. The inversion jump $\Delta\theta$, lapse rate $\gamma$, reference potential temperature $\theta_0$, inversion width $\Delta h$, inversion height $H$, friction velocity $u^*$, geostrophic wind $G$ and geostrophic wind angle $\phi_G$ are also reported for each case in Table A1. The first five parameters have been obtained by fitting the potential temperature profile averaged between $100\,000$ s to $120\,000$ s and $100\,000$ s to $140\,000$ s for the AGW-resolved and AGW-modeled cases, respectively, using the Rampanelli and Zardi (2004) model. Since all simulations employ the potential temperature controller described in Stipa et al. (2024a), these quantities exactly match in all cases, which is further confirmed by the potential temperature profiles of Figure A1. The shear stress profile also agrees almost perfectly and all cases are characterized by the same final friction velocity $u^*$. Conversely, the wind speed magnitude and wind angle shows some minor differences, which we attribute to the use of the geostrophic damping technique.

| domain size [km] | $\Delta\theta$ [K] | $\gamma$ [K km$^{-1}$] | $\theta_0$ [K] | $\Delta h$ [m] | $H$ [m] | $u^*$ [m s$^{-1}$] | $G$ [m s$^{-1}$] | $\phi_G$ [°] |
|---|---|---|---|---|---|---|---|---|
| $6 \times 3 \times 1$ | 7.312 | 1 | 300.0 | 98.1 | 500.0 | 0.43 | 10.5 | -24.0 |
| $6 \times 21 \times 1$ | 7.312 | 1 | 300.0 | 98.1 | 500.0 | 0.43 | 10.3 | -23.7 |
| $6 \times 3 \times 1$ | 4.895 | 1 | 300.0 | 95.1 | 500.0 | 0.43 | 10.5 | -23.8 |
| $6 \times 21 \times 1$ | 4.895 | 1 | 300.0 | 95.2 | 500.0 | 0.43 | 10.3 | -23.6 |

**Table A1.** Inversion jump $\Delta\theta$, lapse rate $\gamma$, reference potential temperature $\theta_0$, inversion width $\Delta h$, inversion height $H$, friction velocity $u^*$, geostrophic wind $G$ and geostrophic wind angle $\phi_G$ evaluated from the off-line precursor simulations used for the AGW-resolved and AGW-modeled cases. The first five parameters have been obtained by fitting the potential temperature profile with the Rampanelli and Zardi (2004) model, after averaging in time from $100\,000$ s to $120\,000$ s and from $100\,000$ s to $140\,000$ s for the AGW-resolved ($6 \times 3 \times 1$ km domain) and AGW-modeled ($6 \times 21 \times 1$ km domain) cases, respectively.

Geostrophic damping is used to eliminate inertial oscillations produced by a geostrophic momentum imbalance when initializing the flow without any knowledge about the geostrophic wind. As explained in Stipa et al. (2024a), this situation occurs when one tries to control the horizontally-averaged wind velocity somewhere inside the boundary layer. In this case, it is impossible to know *a-priori* what the geostrophic wind will be, and $G$ — required to apply the geostrophic damping action — has to be retrieved by inverting the equations for the geostrophic balance, using the pressure gradient calculated by the velocity controller. In turn, the pressure gradient is calculated by horizontally averaging the wind components at each iteration

during the simulation and thus it is expected that the averaging procedure yields slightly different values when using domains of different size. However, the difference in the final values of geostrophic wind between the small and the large domains is only about $0.2 \, \text{m s}^{-1}$, which is acceptable in the context of the present study.

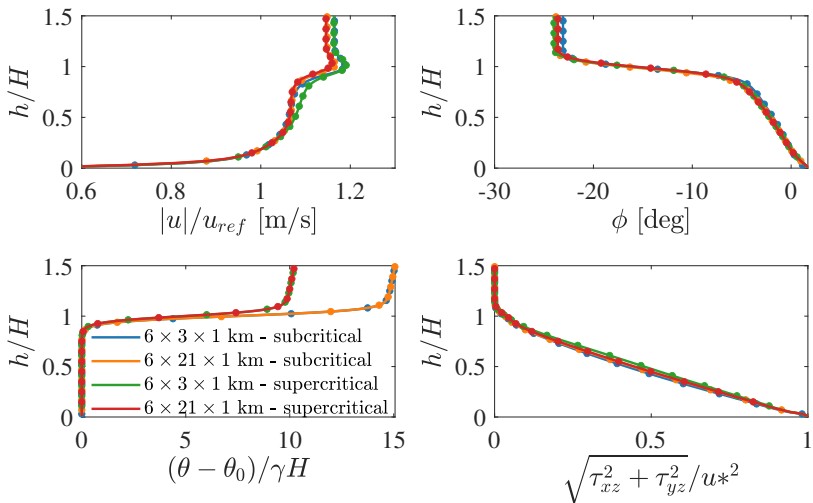

**Figure A1.** Horizontally and time averaged wind speed magnitude (top-left), wind angle (top-right), shear stress (bottom-right) and potential temperature (bottom-left) evaluated from the off-line precursor simulations used for the AGW-resolved and AGW-modeled cases. Time averaging has been performed from $100\,000$ s to $120\,000$ s and from $100\,000$ s to $140\,000$ s for the AGW-resolved ($6 \times 3 \times 1$ km domain) and AGW-modeled ($6 \times 21 \times 1$ km domain) cases, respectively.

## Appendix B: Effect of Atmospheric Turbulence on Thrust and Power Averages

As mentioned in Section 3, although precursor simulations for the AGW-resolved and AGW-modeled cases shared the same inputs parameters, they run on domains of different size. As a consequence, while simulations corresponding to the same CNBL conditions produced almost identical horizontally-averaged fields (see Appendix A), these feature different realizations of the time-resolved turbulent field. Although this does not represent an issue for the small eddies, large turbulent structures may change the freestream velocity obtained by averaging over a time window that is comparable with their size. This is evident from Figure B1b, which reports the instantaneous wind, averaged among the first row turbines, for the AGW-resolved and AGW-modeled approaches corresponding to both subcritical and supercritical conditions. When averaging time histories of e.g. turbine power or thrust, this effect can introduce a consistent bias as these quantities depend on the cube and square of velocity, respectively. Unfortunately, for a time window of the order of the one available in the AGW-resolved simulations ($15\,000$ s), such effect introduces a variability in wind farm thrust and power that is comparable with the effect of blockage. This is shown in Figure B1a, where we report the time history of the velocity sampled at the domain inlet and close to the wind turbine located at the first row center, for both the subcritical and supercritical AGW-modeled simulations. As can be

noticed, the two curves are vertically shifted due to blockage effects. However, even though 15 000 s can be considered a large averaging window, the variations in average velocity obtained by hypothetically shifting this window in time are expected to be comparable with, if not larger than, the vertical shift produced by blockage.

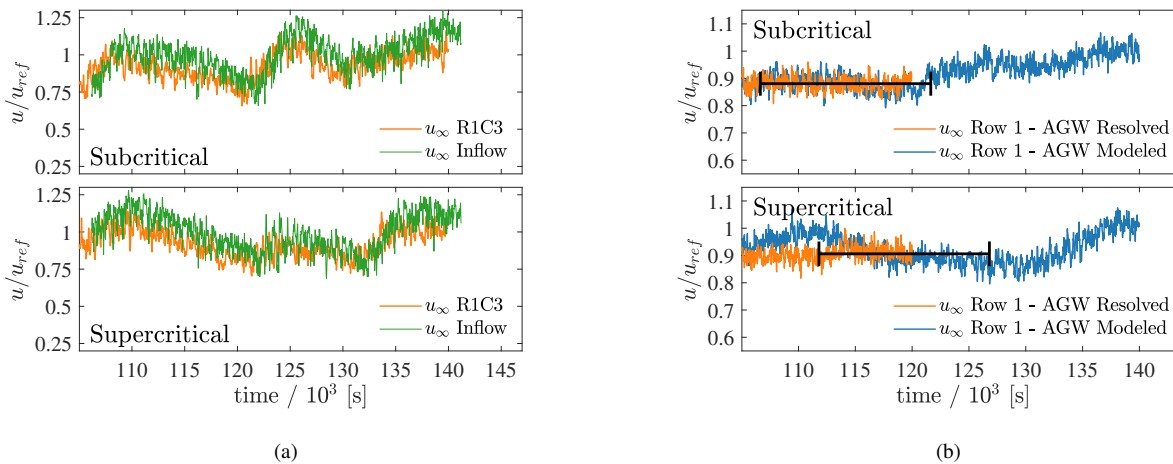

(a)                                             (b)

**Figure B1.** (a) Hub-height wind speed at the domain inlet (green) and as sampled by the wind turbine located in the middle of the first row (orange). Data correspond to the subcritical and supercritical AGW-modeled simulations. (b) Wind speed averaged among the first row wind turbines for the AGW-resolved (orange) and AGW-modeled (blue) cases. Data corresponding to both subcritical and supercritical conditions are shown. The black bar indicates the time window used to average turbine data in the AGW-modeled simulations, when these are compared against the AGW-resolved simulations.

     For this reason, when comparing turbine power and thrust between AGW-modeled and AGW-resolved methods under the same CNBL conditions, we chose the averaging window for the former case as follows. First, we ensure that the same window

is used for both cases, i.e. 15 000 s, corresponding to the entirety of the data available from the AGW-resolved analyses. Then, we shift the averaging window in the AGW-modeled cases until the freestream velocity averaged among the first row turbines matches the same quantity obtained from the AGW-resolved simulation. The averaging window resulting from this approach is reported in black in Figure B1b, for both subcritical and supercritical conditions. Finally, turbine thrust and power from the AGW-modeled cases are averaged over this window, ensuring that the wind farm sees the same inflow velocity in the two cases.

We emphasize that this procedure is only applied when looking at turbine data, while the flow variables are always averaged throughout the entire simulation, i.e. from 105 000 to 120 000 s for the AGW-resolved and from 105 000 s to 140 000 s for the AGW-modeled cases. Although it is true that the approach described above likely forces the first row average power to match between AGW-resolved and AGW-modeled cases characterized by the same CNBL conditions, it should be recognized that it also allows thrust and power distributions to vary in the remaining rows. As a consequence, this does not impair our ability to

assess the accuracy of the AGW-modeled technique in capturing AGW effects on the row-by-row power production.

*Code availability.* TOSCA is available at https://doi.org/10.17605/OSF.IO/Q4VAF (Stipa et al., 2023).

*Data availability.* The dataset can be made available from the authors upon request.

*Author contributions.* Conceptualization, SS, MAK, DA, JB; methodology, SS, MAK; software, SS; validation, SS; formal analysis, SS; investigation, SS, MAK; computational resources, JB; data curation, SS; writing–original draft preparation, SS, MAK; writing–review and editing, JB, DA; visualization, SS; supervision, JB, DA; project administration, JB; funding acquisition, JB. All authors have read and agreed to the published version of the manuscript.

*Competing interests.* The contact author has declared that none of the authors has any competing interests.

*Acknowledgements.* Computational resources provided by the Digital Research Alliance of Canada and Advanced Research Computing at the University of British Columbia are gratefully acknowledged.

*Financial support.* This research has been supported by the Natural Sciences and Engineering Research Council of Canada (grant no. 556326).

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
