# Peer review of "An LES Model for Wind Farm-Induced Atmospheric Gravity Wave Effects Inside Conventionally Neutral Boundary Layers."

_Wind Energy Science, 2023_

## Referee Comment (RC1)

In this article, the authors develop a new methodology for simulating wind-farm induced gravity waves in an LES framework at a considerably lower computational cost. This technique is based on the fact that gravity-wave induced pressure perturbations are dependent on the vertical displacement of the capping inversion. Therefore, the multi-scale coupled (MSC) model is adopted to compute this displacement, which is given as input to TOSCA, a LES solver. The latter only simulates the flow in the atmospheric boundary layer, as gravity-wave effects are taken into account by the MSC model. The results obtained with this technique, named AGW-modelled, are compared against LESs that resolve the full domain (i.e. including the free atmosphere), which typically necessitate lots of computational resources. The comparison shows that the AGW-modelled velocity and pressure-perturbation profiles are in good agreement with the ones predicted by the AGW-resolved simulations. Moreover, the extreme case of a rigid lid is also explored by fixing the vertical extent of the domain to the height of the capping inversion.

In my opinion, the manuscript is generally well written. Moreover, the ideas are clearly presented, which makes the reasonings easy to follow in most of the sections. I also believe that this article is of interest to the wind energy community as it shows a new methodology for investigating wind-farm operations in conventionally neutral boundary layers which considerably reduces the computational cost of LES while maintaining a good level of accuracy. In the following, I list comments on how the paper may be improved.

Scientific comments/questions

1. Abstract: the authors mention that this new technique demands less than 15% of the computational resources than traditional methods, which is an impressive achievement. However, this technique also reduces the accuracy of an LES to the one of the MSC model. I think this is quite an important limitation, which should be mentioned.

2. Line 34: the authors mention that AGWs have "extremely large spatial scales". I think the reader would benefit if a number in terms of km would be provided, as the word "extremely" can be subjective. I suggest trying to limit the use of this word throughout the manuscript (e.g. see lines 35 and 51).

3. Line 37: I would also refer to some earlier works about AGW excited by hills/mountains (e.g. Klemp and Lilly 1977, Teixeira 2014).

4. Line 54-67: the authors describe the Rayleigh damping layer and fringe region technique in this paragraph. However, the descriptions are mixed so that it is easy to confuse the role of the two techniques. I would propose to re-write this paragraph explaining first the Rayleigh damping technique, for instance, and afterwards the working principle and use of the fringe region technique.

5. Line 78: the authors define the rigid-lid approximation as a case with strong free atmosphere stability. However, this could also be seen as a case with a very strong (practically infinite)

capping-inversion strength. To avoid confusion, I would define the rigid-lid case as an approximation for a case with very strong thermal stratification above the atmospheric boundary layer.

6. Introduction: I suggest to include the general aim of the article in the first paragraph of the introduction. This will give the reader a hint about why AGW numerical models and boundary conditions are described in this section.

7. Line 107: I would mention this in the introduction, i.e. that the article only deals with CNBLs.

8. Section 2.2: the proposed technique uses the MSC model to predict the capping-inversion vertical displacement (eta) and only simulate the flow within the ABL with LES. Hence, it is implicitly assumed that $H>H1$, with $H1=2*z\_hub$. However, $H1$ values are getting closer to heights at which the capping inversion is typically located (for instance, the Vestas V172-7.2MW turbine has a hub height of 199 m, a figure that may increase in the coming years) so that cases where the ratio $H/H1 \approx 1$ are realistic. In the current work, the authors fix $H/H1$ to 2.77 and do not discuss this further. However, I believe that the performance of the new technique is sensitive to the $H/H1$ ratio. Would it be possible to add a few comments and/or simulations on what would happen when $H/H1 \approx 1$ ? Also, how would this technique deal with cases where $H/H1 < 1$? If this is a limitation, it should be reported in the text.

9. Line 184: the authors mention that "Then, the vertical displacement is linearly distributed to the underlying cells, deforming the mesh before starting the simulation". This passage is not clear to me. Would it be possible to explain in more detail?

10. Line 193: in the LES framework, the vertical displacement of the capping inversion generates a cold anomaly, which in turn results in pressure perturbation. However, the authors mention that in CNBLs, the potential-temperature equation can be left out in the AGW-modelled simulations. Hence, are the pressure perturbations solely caused by the flow convergence/divergence caused by the irregular top edge of the main domain? How is the buoyancy term computed in the vertical momentum equation? Or is it neglected? I would appreciate a more detailed explanation.

11. Line 194: the authors mention that "This condition is only violated very close to the top boundary, where discrepancies in turbulent fluctuations produced by the absence of stability and by the physical boundary are deemed acceptable as they happen away from the wind farm". I would note that this affirmation does not hold for low $H/H1$ ratios (as mentioned in comment 8).

12. Section 3.2: if I understand correctly, the inflow data used for the AGW resolved and modelled simulations are computed with two different precursor techniques. Hence, differences in the wind-farm simulation results between the two techniques could be also attributed to this difference in inflow conditions. Why the authors do not drive the AGW-modelled simulations with inflow sections taken from the precursor simulations used for the AGW-resolved simulations?

13. Line 264: the authors mention that "we used the velocity inflow data of the subcritical case to prescribe an inlet for the rigid lid". Is the velocity profile in the precursor domain for sub- and supercritical cases equal? Showing some vertical profiles of the precursor simulation would be beneficial.

14. Figure 2: I suggest to center the colorbar around the zero value (so that the background color is white).

15. Line 274: the authors mention that "the AGW-modeling technique requires a domain that is more than 85% smaller compared to the AGW-resolving approach". In which terms? Number of cells? Does this number also consider the precursor domain (for instance, the AGW-modelled simulations use a precursor domain 7 times bigger in the y-direction)?

16. Figure 3: How would you explain the differences between AGW-modelled simulations and the MSC model? Are those due to the simplifications made in the MSC model (for instance, linearity, absence of resolved turbulence, etc..)?

17. Figure 3: I suggest extending the caption of this figure, mentioning for instance that the profiles shown are averaged in height between H1 and H and along the wind-farm width. This comment extends to the whole manuscript since I feel that the combination of figure and caption is often not self-explanatory.

18. Line 304: the domain in Figure 3 is too small to appreciate this behaviour.

19. Line 313: at which height are the profiles taken? I suggest mentioning it in the manuscript.

20. Figure 4: the match in terms of velocity is really good. This makes it hard to spot differences between AGW-modelled and resolved simulations in Figure 4, which obviously is a good sign. Could be an idea to also plot the relative error for both cases? This will allow the reader to easily understand where the two methods differ the most.

21. Table 3: Would it be possible to include the non-local, wake and farm efficiency values in this table (as defined in Lanzilao and Meyers (2024), for instance)? The total farm power (and farm efficiency) in the supercritical case is almost identical in the AGW-M and AGW-R cases. However, I expect some differences in non-local and wake efficiencies.

22. Section 4.2: I'm assuming that the rigid lid is located at H=500 m. Is this correct? I suggest to explicitly mention the vertical extent of the domain in the text.

23. Section 4.2: the pressure build-up in the rigid-lid case is solely attributed to flow confinement. However, even in neutral conditions, the cumulative turbine induction generates a pressure build-up and consequently, a flow slow down (typically much lower in values than the one observed in the presence of thermal stratification). Therefore, I would rephrase this sentence

and/or find an alternative method to evaluate the flow blockage solely induced by the flow confinement.

24. Line 350: Which is the MSC setup used to generate Figure 8? In the text, only the capping-inversion strength and free lapse rate are mentioned. In general, I would appreciate more details on the simulation setup, so that it would get easier and more intuitive to reproduce the results.

25. Figure 8: it would be interesting to split this figure into three panels, reporting the sensitivity of the relative error based on non-local, wake and farm efficiency to the capping-inversion strength and free lapse rate. I suggest this because at times the total farm power of two simulations can be almost identical, even though the power trends are very different (two behaviours that cancel out).

26. Table 4: the relative error remains positive for all cases when the MSC model is used. However, in the LES case, the error becomes negative for the subcritical case. Any idea about why this occurs?

27. Line 372-374: the fringe region is adopted in pseudo-spectral (or fully spectral) flow solvers to impose the inflow conditions. The presence of gravity waves does not imply the use of a fringe region, as inflow-outflow boundary conditions can be adopted (although the implementation is not trivial).

28. Line 385: As a future work, I would also suggest further validation of this technique, as it has been only applied to two idealized cases. For instance, Lanzilao and Meyers (2024) performed 40 LESs in different atmospheric conditions, for which the displacement of the capping inversion is computed. This comparison could offer further insights into the performance of the proposed method.

29. Appendix A: this construction looks quite artificial to me. From my point of view, it would be easier to drive the main domain using the same precursor simulation for both the AGW-modelled and AGW-resolved simulations. Is there a particular reason why the authors decided to not pursue this option? It would eliminate the need for the process described in this appendix together with ensuring equal inflow conditions in both cases.

Technical comments

1. Line 113: replace "If one wishes to resolve AGW within LES" with "When simulating AGWs in an LES framework" or similar

2. line 205: used -> use

3. Line 368: conventionally neutral boundary layers -> CNBLs

---

## Referee Comment (RC2)

**Review: An LES Model for Wind Farm-Induced Atmospheric Gravity Wave Effects Inside Conventionally Neutral Boundary Layers**

**Summary**

The authors propose a new computational approach to include the impact of wind far-generated gravity wave-effects in LES of conventionally neutral boundary layers that is more computationally efficient than other approaches used in the literature. The authors use results from a multi-layer model of the atmosphere to deform the inversion layer in the LES and account for the effect of wind far-triggered gravity waves on the flow. The authors show proof of concept for two flow cases, a subcritical and supercritical flow. Their new modeling approach provides a realistic representation of gravity wave effects within the boundary layer when compared to simulations that follow common practices. In general, the manuscript is well written, and the results are sound. However, I have some comments that should be addressed before publication. Mainly, the authors make strong statements about the causes of blockage that should be revisited. Based on the results presented in this paper, it seems gravity waves play a secondary role in velocity reductions upstream of the wind farm. However, there is a lot of emphasis throughout the paper on gravity waves causing most of the global blockage effect.

**Major Comments:**

1. Global blockage effects: The authors are drawing strong conclusions on the mechanisms that cause the global blockage effect. They attribute the velocity deceleration upstream of the wind farm either to a gravity wave-induced pressure gradient or to flow confinement (e.g., Lines 41-43, Lines 185-187, Lines 289-291, Lines 344-345, Lines 388-389). I agree that flow confinement and gravity waves may play a role in these cases; however, the deceleration of the wind upstream of the wind farm can also be due to other mechanisms that are likely present in these simulations but that are not discussed here (Bleeg and Montavon, 2022; Sanchez Gomez et al., 2023). In fact, the authors clearly show that other mechanisms (i.e., not gravity wave-induced velocity decelerations) are responsible for more than 50% of the velocity deceleration upstream of the wind farm and gravity-wave-induced blockage is secondary (Figure 6).

2. Rigid-lid approximation: The authors use the rigid-lid approximation throughout the manuscript; however, it is not clear what is the purpose of using such a simplified and unrealistic modeling approach. In Lines 83-85, the authors suggest the rigid-lid approximation may be useful for use in engineering parameterizations. What do the authors mean by engineering parameterizations? Also, the rigid-lid approximation is tested here neutral boundary layer flow, which is unrealistic compared to the atmospheric boundary layer. For example, Bleeg and Montavon (2022) show that neglecting the temperature stratification in the capping inversion and troposphere misrepresents the blockage effect.

**Minor Comments:**

1. Line 153-154: Why are the wind farm and upper layer characterized by the same background velocity? This assumption virtually discards the effect from shear and the large gradients associated with the atmospheric surface layer.

2. Figure 2: The divergent color map is not centered at 0, making it very difficult to distinguish between positive and negative inversion displacements.

3. Lines 299-300: I would argue that the AGW-modeled and AGW-resolved approaches do not predict almost the same pressure perturbation for the subcritical case (Figure 3a). Differences in the pressure perturbation field between the AGW resolved and modeled approaches are at least on the order of 10% upstream of the wind farm.

4. Lines 307-312: The differences upstream of the wind farm are just as large (or larger) than the differences at the domain outflow. However, the hypothesis presented by the authors does not address these differences. The flow upstream of the wind farm is outside and downstream of the fringe region and these differences are still large.

5. Lines 344-345: The authors conclude that flow confinement is responsible for blockage to a lesser extent than gravity waves. However, Figure 6 clearly shows that the velocity deceleration with gravity waves is less than twice as large as the deceleration in the rigid-lid simulations. Thus, it seems flow deceleration from gravity wave-induced pressure gradients is not the main cause for blockage in these simulations. Also, I would argue that flow confinement is not the only cause for blockage in the rigid-lid case.

6. The authors mention that the LES domain should extend to one or more wavelengths in each direction (Line 113). However, extending the LES above ~10-12 km in the atmosphere means you are performing simulations above the tropopause, where the temperature stratification is very different from the constant lapse rate assumed within the troposphere. Is gravity wave propagation sensitive to having multiple thermally stratified layers like in the atmosphere compared to a single constant lapse rate? This might be out of the scope of the paper but is something to consider.

**References**

Bleeg, J. and Montavon, C.: Blockage effects in a single row of wind turbines, J. Phys.: Conf. Ser., 2265, 022001, https://doi.org/10.1088/1742-6596/2265/2/022001, 2022.

Sanchez Gomez, M., Lundquist, J. K., Mirocha, J. D., and Arthur, R. S.: Investigating the physical mechanisms that modify wind plant blockage in stable boundary layers, Wind Energ. Sci., 8, 1049–1069, https://doi.org/10.5194/wes-8-1049-2023, 2023.

---

## Author Comment (AC1)

**University of British Columbia**

**UBCO-UL NSERC Alliance Grant "Reduced-Order Models of Wind Farm Induction and Far-Field Wake Recovery"**

**Response to Reviewer 1**

Exec. S. Stipa - March 7, 2024

We would like to thank the reviewer for the time dedicated to revising the paper. We proceed with answering and clarifying, where possible, their comments.

Our response, denoted in black, is shown below, while the reviewer's comments are denoted in blue. Please refer to the track changes document for a detailed overview of the changes made to the manuscript.

Abstract: the authors mention that this new technique demands less than 15% of the computational resources than traditional methods, which is an impressive achievement. However, this technique also reduces the accuracy of an LES to the one of the MSC model. I think this is quite an important limitation, which should be mentioned.

This aspect has been added to the abstract (see line 15 of the track changes document).

Line 34: the authors mention that AGWs have "extremely large spatial scales". I think the reader would benefit if a number in terms of km would be provided, as the word "extremely" can be subjective. I suggest trying to limit the use of this word throughout the manuscript (e.g. see lines 35 and 51).

The word "extremely" has been omitted and more specific details have been added (see lines 40-42 and 171-172 of the track changes manuscript).

Line 37: I would also refer to some earlier works about AGW excited by hills/mountains (e.g. Klemp and Lilly 1977, Teixeira 2014).

The reference Teixeira (2014) has been added at line 37 of the track changes manuscript, while the work from Klemp and Lilly (1978) has been cited later at line 40 to introduce the problem of wave reflection.

Lines 54-67: the authors describe the Rayleigh damping layer and fringe region technique in this paragraph. However, the descriptions are mixed so that it is easy to confuse the role of the two techniques. I would propose to re-write this paragraph explaining first the Rayleigh damping technique, for instance, and afterwards the working principle and use of the fringe region technique.

The paragraph has been almost completely rewritten following the reviewer's comment to ensure that the Rayleigh damping layer and fringe layer are distinctly described. Please refer to the track changes document.

Line 78: the authors define the rigid-lid approximation as a case with strong free atmosphere stability. However, this could also be seen as a case with a very strong (practically infinite) capping-inversion strength. To avoid confusion, I would define the rigid-lid case as an approximation for a case with very strong thermal stratification above the atmospheric boundary layer.

The comment has been implemented in the revised manuscript. Please see lines 24 and 122 of the track changes document.

Introduction: I suggest to include the general aim of the article in the first paragraph of the introduction. This will give the reader a hint about why AGW numerical models and boundary conditions are described in this section.

The comment has been implemented in the revised manuscript. Please see lines 45-47 of the track changes document.

Line 107: I would mention this in the introduction, i.e. that the article only deals with CNBLs.

The comment has been implemented in the revised manuscript. Please see lines 143-144 of the track changes document.

Section 2.2: the proposed technique uses the MSC model to predict the capping-inversion vertical displacement ($\eta$) and only simulate the flow within the ABL with LES. Hence, it is implicitly assumed that $H > H_1$, with $H_1 = 2z_{\text{hub}}$. However, $H_1$ values are getting closer to heights at which the capping inversion is typically located (for instance, the Vestas V172-7.2MW turbine has a hub height of 199 m, a figure that may increase in the coming years) so that cases where the ratio $H/H_1 \approx 1$ are realistic. In the current work, the authors fix $H/H_1$ to 2.77 and do not discuss this further. However, I believe that the performance of the new technique is sensitive to the $H/H_1$ ratio. Would it be possible to add a few comments and/or simulations on what would happen when $H/H_1 \approx 1$ ? Also, how would this technique deal with cases where $H/H_1 < 1$? If this is a limitation, it should be reported in the text.

This comment has been now addressed in the manuscript as follows, between lines 349 and 363 of the track changes document). The case where $H/H_1 \approx 1$ "corresponds to a situation where the turbine top tip almost pierces the inversion layer, with consequent disappearance of the upper layer. Devesse et al. (2023) developed an alternative strategy to the one used in the MSC model to couple the 3LM of Allaerts and Meyers (2019) and the Bastankhah and Porté-Agel (2014) wake model, which also uses the 3LM to address AGW effects. When validating this new model against wind farm LES characterized by $H = 150, 300, 500$ and 1000 m and $h_{hub} = 119$ (Lanzilao and Meyers, 2023), the authors excluded those LES cases with $H/H_1 = 0.63$ ($H = 150$ m). Among the remaining cases, the model showed the highest deviation from the LES when $H/H_1 = 1.26$ ($H = 300$ m). As also the MSC model uses the 3LM to model AGW effects, these results suggest that the MSC model will loose accuracy when $H/H_1 \lesssim 1.5$. In the present manuscript, the dependency of the proposed technique to the ratio $H/H_1$ is not investigated and this number is fixed to 2.78. We highlight that this is a limitation of the MSC model used to compute $\eta$. If $\eta$ could be evaluated by different means (e.g. with a coarser AGW-resolved LES employing a simple canopy model) at a height located above the inversion layer, the AGW modeling approach could be used for small $H/H_1$ ratios by placing the upper boundary a few hundreds meters into the free atmosphere and by including the potential temperature transport equation."

Line 184: the authors mention that "Then, the vertical displacement is linearly distributed to the underlying cells, deforming the mesh before starting the simulation". This passage is not clear to me. Would it be possible to explain in more detail?

This aspect has been clarified in the manuscript. Please see lines 322 to 324 of the track changes document. Specifically, the upper boundary initially located at $H$ is vertically displaced according to $\eta$ before starting the simulation, following which it remains fixed, as the applied $\eta$ corresponds to the steady state solution obtained with the MSC. Then, the vertical displacement applied to the top boundary is linearly distributed to the underlying cells. This means that, at each horizontal location, the first cell away from the bottom wall is not displaced at all, while the top cell moves vertically by $\eta$. In between, the cells are vertically displaced by a distance $\Delta d$ that is calculated based on their distance from the wall as $\Delta d(\mathbf{x}) = z/L_z \cdot \eta(x,y)$, where $L_z$ is the local vertical domain size and $\Delta d$ is the vertical displacement at $\mathbf{x}$.

Line 193: in the LES framework, the vertical displacement of the capping inversion generates a cold anomaly, which in turn results in pressure perturbation. However, the authors mention that in CNBLs, the potential-temperature equation can be left out in the AGW-modelled simulations. Hence, are the pressure perturbations solely caused by the flow convergence/divergence caused by the irregular top edge of the main domain? How is the buoyancy term computed in the vertical momentum equation? Or is it neglected? I would appreciate a more detailed explanation.

On lines 140-144 and 341-342 of the track changes document, we state that under CNBLs there is no need to solve the potential temperature advection equation, as the flow is neutral everywhere except close to the top boundary. For those conditions where $H/H_1 \approx 1$, the upper boundary should be moved a few hundreds meters into the free atmosphere and so potential temperature must be solved and $\eta$ cannot be calculated with the MSC model anymore (though a coarse LES using a canopy model might be used). This aspect is addressed at the end of Section 2.2.

Moreover, a more detailed explanation on the relation between flow convergence/divergence and AGWs is provided between lines 328-333 of the track changes manuscript, when talking about the rigid lid. To summarize, buoyancy is not required to capture AGW effects inside the ABL, as these are given by flow divergence/convergence of the ABL top. In fact, the pressure disturbance produced within the ABL by AGWs in the free atmosphere has to coincide with the pressure produced by flow convergence/divergence, otherwise the governing equations (Equations 9 and 10 of the revised manuscript) are not satisfied. In particular, there is a unique $\eta$ that satisfies this condition, which is the one that we impose using the MSC model. This whole reasoning is the backbone of Section 2.2 and it is shown using the simple model derived by simplifying the 3LM model of Allaerts and Meyers (2019).

Line 194: the authors mention that "This condition is only violated very close to the top boundary, where discrepancies in turbulent fluctuations produced by the absence of stability and by the physical boundary are deemed acceptable as they happen away from the wind farm". I would note that this affirmation does not hold for low $H/H_1$ ratios (as mentioned in comment 8).

We agree with the reviewer, but in this case the main problem would be not being able to use the MSC model to compute $\eta$. To extend our approach to low $H/H_1$ ratios, we would advocate using a coarse LES that employs a canopy model to run a computationally cheaper AGW-resolved simulation. Under those conditions, the domain in the AGW-modeling method can be truncated a few hundred meters into the free atmosphere, instead of at $H$, as the streamline displacement is available here from the AGW-resolved LES. Of course, potential temperature transport has to be retained in this case even if a CNBL is simulated. This suggestion is given at lines 359-363 of the track changes document.

Section 3.2: if I understand correctly, the inflow data used for the AGW resolved and modelled simulations are computed with two different precursor techniques. Hence, differences in the wind-farm simulation results between the two techniques could be also attributed to this difference in inflow conditions. Why the authors do not drive the AGW-modelled simulations with inflow sections taken from the precursor simulations used for the AGW-resolved simulations?

We agree with the reviewer that this would have been the best approach. However, the inflow data used for the AGW-resolved cases is not available as it was generated at runtime during the simulations (these employed a concurrent precursor method) and not saved to slices. Hence, the approach followed in the manuscript is arguably the best alternative. A supporting rationale has been added to the paper (lines 431-434 of the track changes document).

Line 264: the authors mention that "we used the velocity inflow data of the subcritical case to prescribe an inlet for the rigid lid". Is the velocity profile in the precursor domain for sub- and supercritical cases equal? Showing some vertical profiles of the precursor simulation would be beneficial.

The analysis required by the reviewer has been added to the revised manuscript in Appendix A.

Figure 2: I suggest to center the colorbar around the zero value (so that the background color is white).

The reviewer's comment has been implemented in the revised manuscript.

Line 274: the authors mention that "the AGW-modelling technique requires a domain that is more than 85% smaller compared to the AGW-resolving approach". In which terms? Number of cells? Does this number also consider the precursor domain (for instance, the AGW-modeled simulations use a precursor domain 7 times bigger in the y-direction)?

This comment has been addressed in more detail in the revised manuscript (see lines 479-482 of the track changes document).

Figure 3: How would you explain the differences between AGW-modelled simulations and the MSC model? Are those due to the simplifications made in the MSC model (for instance, linearity, absence of resolved turbulence, etc..)?

We explain the differences as follows. The AGW-modeled and MSC model feature the exact same $\eta$, but a different level of fidelity inside the boundary layer. Hence, the same $\eta$ does not lead to identical pressure and velocity perturbations. Conversely, the AGW-modeled and AGW-resolved cases use the same model inside the ABL, but $\eta$ is slightly different, as it comes from the MSC model in the former and it is resolved in the latter. As a consequence, mass and momentum conservation show some differences in the perturbation velocity and pressure. This explanation has been added to the revised manuscript (see lines 518-529 of the track changes document).

Figure 3: I suggest extending the caption of this figure, mentioning for instance that the profiles shown are averaged in height between H1 and H and along the wind-farm width. This comment extends to the whole manuscript since I feel that the combination of figure and caption is often not self-explanatory.

The reviewer's comment has been addressed in the revised manuscript and further extended to all figure captions. Please refer to the track changes document.

Line 304: the domain in Figure 3 is too small to appreciate this behavior.

We have rephrased by pointing at Figure 3 in the revised manuscript, which corresponds to Figure 2 of the original manuscript (i.e. the entire $\eta$ fields for the subcritical and supercritical conditions).

Line 313: at which height are the profiles taken? I suggest mentioning it in the manuscript.

They are taken at the hub height. The reviewer's comment has been implemented in the revised manuscript (see line 537 of the track changes document).

Figure 4: the match in terms of velocity is really good. This makes it hard to spot differences between AGW-modelled and resolved simulations in Figure 4, which obviously is a good sign. Could be an idea to also plot the relative error for both cases? This will allow the reader to easily understand where the two methods differ the most.

The reviewer's suggestion has been implemented in the revised manuscript.

Table 3: Would it be possible to include the non-local, wake and farm efficiency values in this table (as defined in Lanzilao and Meyers (2024), for instance)? The total farm power (and farm efficiency) in the supercritical case is almost identical in the AGW-M and AGW-R cases. However, I expect some differences in non-local and wake efficiencies.

The reviewer's comment has been addressed. However, since the inflow data relative to the AGW-resolved simulations is not available, it is impossible to conduct isolated with turbine simulations to compute $P_\infty$, as done in Lanzilao and Meyers (2023). For this reason, in order to compute the efficiency, we used the data from Appendix B of Stipa et al. (2023), which are evaluated with uniform inflow and in absence of turbulence. We agree that this would lead to values of $\eta_{nnl}$ and $\eta_{tot}$ that are different from the actual figures, but since the same procedure to compute $P_\infty$ has been used for all the entries of Table 4 of the revised manuscript, comparisons can still be drawn (lines 561-586 of the track changes document). In particular, the ability of the model to capture the underlying physics is confirmed by noticing that the AGW-modeled and AGW-resolved simulations lead to the same conclusions regarding which case is characterized by the highest wake efficiency, blockage effect and total wind far power.

Section 4.2: I'm assuming that the rigid lid is located at $H = 500$ m. Is this correct? I suggest to explicitly mention the vertical extent of the domain in the text.

Correct. This information has been added (see line 595 of the track changes document).

Section 4.2: the pressure build-up in the rigid-lid case is solely attributed to flow confinement. However, even in neutral conditions, the cumulative turbine induction generates a pressure build-up and consequently, a flow slow down (typically much lower in values than the one observed in the presence of thermal stratification). Therefore, I would rephrase this sentence and/or find an alternative method to evaluate the flow blockage solely induced by the flow confinement.

This section has been heavily modified (please see the track changes document). To specifically address the reviewer's comment, we would like to highlight the statement added at lines 605-609 of the track changes document. In particular, global blockage effects is always due to flow confinement, which is an alternative way of referring to the AGW-induced pressure gradient. In fact, the two are uniquely related, as explained in Section 2.2. Hence, in the rigid-lid global blockage is generated in the exact same manner as in the full AGW solution, with the only difference being that flow confinement is approximated to that produced when $\eta = 0$. This implies that the flow is horizontally divergence free in the rigid lid (i.e. on wall-parallel planes), while continuity is satisfied on pliant surfaces defined by $\eta$ in the full AGW solution (i.e. surfaces coincident with the local vertical streamline displacement). Notably, both induce global blockage due to flow confinement or, alternatively, to stability effects above $H$, but the rigid lid corresponds to the limiting case where $\Delta\theta \to \infty$ and/or $\gamma \to \infty$.

Line 350: Which is the MSC setup used to generate Figure 8? In the text, only the capping-inversion strength and free lapse rate are mentioned. In general, I would appreciate more details on the simulation setup, so that it would get easier and more intuitive to reproduce the results.

The reviewer's request has been implemented throughout the revised paper (see for instance Table 3 of the revised manuscript).

Figure 8: it would be interesting to split this figure into three panels, reporting the sensitivity of the relative error based on non-local, wake and farm efficiency to the capping-inversion strength and free lapse rate. I suggest this because at times the total farm power of two simulations can be almost identical, even though the power trends are very different (two behaviors that cancel out).

The reviewer's request has been addressed in the revised manuscript. In particular, instead of computing the error between the different models, 1D parametric analyses have been conducted (Figures 9 and 10), where the different approaches are directly (and more visually) compared. Moreover, Figure 8 of the old manuscript has been removed and substituted with the sensitivity of $\eta_{nnl}$, $\eta_w$ and $\eta_{tot}$ to the parameters $\Delta\theta$ and $\gamma$. The error when these are estimated using the rigid lid approximation can still be well appreciated

from Figures 9 and 10.

Table 4: the relative error remains positive for all cases when the MSC model is used. However, in the LES case, the error becomes negative for the subcritical case. Any idea about why this occurs?

The sensitivity study has been extended, and there are indeed conditions where the rigid lid approximation performs slightly worse than the full AGW solution. In fact, it appears that $\eta_{tot}$ approaches the rigid lid solution from above instead of below. The cross-over point occurs, according to the MSC model, around $\Delta\theta = 10$ K. This is higher than the value used in the subcritical LES conditions, where $\eta_{tot}$ for the subcritical case is already higher than the rigid lid at 7.312 K. Unfortunately, while the crossover of the full AGW solution over the rigid lid seems to be predicted by both the MSC model and the LES, we do not have a clear explanation regarding the difference in the value of $\Delta\theta$ at which such crossover is observed.

Line 372-374: the fringe region is adopted in pseudo-spectral (or fully spectral) flow solvers to impose the inflow conditions. The presence of gravity waves does not imply the use of a fringe region, as inflow-outflow boundary conditions can be adopted (although the implementation is not trivial).

This aspect has been addressed in Section 2.1. And the sentence mentioned by the reviewer has been corrected by specifically referring to finite volume codes (see line 713).

Line 385: As a future work, I would also suggest further validation of this technique, as it has been only applied to two idealized cases. For instance, Lanzilao and Meyers (2024) performed 40 LESs in different atmospheric conditions, for which the displacement of the capping inversion is computed. This comparison could offer further insights into the performance of the proposed method.

The reviewer's suggestion has been added to the revised manuscript (see lines 727-729 of the track changes document).

Appendix A: this construction looks quite artificial to me. From my point of view, it would be easier to drive the main domain using the same precursor simulation for both the AGW-modelled and AGW-resolved simulations. Is there a particular reason why the authors decided to not pursue this option? It would eliminate the need for the process described in this appendix together with ensuring equal inflow conditions in both cases.

We totally agree with the reviewer. Unfortunately, as previously mentioned, the inflow data for the AGW-resolved simulations (which have been conducted some time ago and already presented in Stipa et al. (2023)) have been generated at runtime and have not been saved to slices in the disk. In this regard, the approach followed in the paper was the only one that allowed us to avoid re-rerunning the AGW-resolved simulations, which we consider an unnecessary use of computational resources in light of the results presented in the paper.

Line 113: replace "If one wishes to resolve AGW within LES" with "When simulating AGWs in an LES framework" or similar.

Corrected (see line 168 of the track changes document).

line 205: used -> use.

Rephrased (see line 370 of the track changes document).

Line 368: conventionally neutral boundary layers -> CNBLs.

Corrected (see line 709 of the track changes document).

**References**

Allaerts, D. and Meyers, J.: Sensitivity and feedback of wind-farm-induced gravity waves, Journal of Fluid Mechanics, 862, 990–1028, https://doi.org/10.1017/jfm.2018.969, 2019.

Bastankhah, M. and Porté-Agel, F.: A new analytical model for wind-turbine wakes, Renewable Energy, 70, 116–123, https://doi.org/https://doi.org/10.1016/j.renene.2014.01.002, special issue on aerodynamics of offshore wind energy systems and wakes, 2014.

Devesse, K., Lanzilao, L., and Meyers, J.: A meso-micro atmospheric perturbation model for wind farm blockage, Submitted to Wind Energy Science Journal, URL https://arxiv.org/abs/2310.18748, 2023.

Klemp, J. and Lilly, D.: Numerical Simulation of Hydrostatic Mountain Waves, J. Atmos. Sci., 35, 78–107, https://doi.org/10.1175/1520-0469(1978)035<0078:NSOHMW>2.0.CO;2, 1978.

Lanzilao, L. and Meyers, J.: A parametric large-eddy simulation study of wind-farm blockage and gravity waves in conventionally neutral boundary layers, 2023.

Stipa, S., Ajay, A., Allaerts, D., and Brinkerhoff, J.: The Multi-Scale Coupled Model: a New Framework Capturing Wind Farm-Atmosphere Interaction and Global Blockage Effects, Wind Energy Science Discussions, 2023, 1–44, https://doi.org/10.5194/wes-2023-75, 2023.

Teixeira, M. A. C.: The physics of orographic gravity wave drag, Frontiers in Physics, 2, https://doi.org/10.3389/fphy.2014.00043, 2014.

---

## Author Comment (AC2)

**University of British Columbia**

**UBCO-UL NSERC Alliance Grant "Reduced-Order Models of Wind Farm Induction and Far-Field Wake Recovery"**

**Response to Reviewer 2**

Exec. S. Stipa - March 7, 2024

We would like to thank the reviewer for the time dedicated to revising the paper. We proceed with answering and clarifying, where possible, the proposed comments.

Our response, denoted in black, is shown below, while the reviewer's comments are denoted in blue. Please refer to the track changes document for a detailed overview of the changes made to the manuscript.

Global blockage effects: The authors are drawing strong conclusions on the mechanisms that cause the global blockage effect. They attribute the velocity deceleration upstream of the wind farm either to a gravity wave-induced pressure gradient or to flow confinement (e.g., Lines 41-43, Lines 185-187, Lines 289-291, Lines 344-345, Lines 388-389). I agree that flow confinement and gravity waves may play a role in these cases; however, the deceleration of the wind upstream of the wind farm can also be due to other mechanisms that are likely present in these simulations but that are not discussed here (Bleeg and Montavon, 2022; Sanchez Gomez et al., 2023). In fact, the authors clearly show that other mechanisms (i.e., not gravity wave-induced velocity deceleration) are responsible for more than 50% of the velocity deceleration upstream of the wind farm and gravity-wave-induced blockage is secondary (Figure 6).

Our statements on the mechanism causing global blockage effects are based on the findings from a number of recent studies that emphasize the critical role played by atmospheric gravity waves (see for example Devesse et al., 2023; Lanzilao and Meyers, 2023; Stipa et al., 2023). However, we acknowledge that these may have not been presented in the clearest way and the manuscript has been heavily revised. Moreover, some additional comments can be made to clarify the specific points addressed by the reviewer.

First, we do not fully agree with the definition of local and global blockage given by Bleeg and Montavon (2022). In particular, they define local blockage as the blockage from immediate neighbors, while global blockage is given by wind-farm-scale blockage effects. In our opinion, this definition does not allow to clearly distinguish between individual turbine induction (and its cumulative effect) and effects related to atmospheric stability. In fact, while local induction acts mainly at the turbine scale, it also has an impact — albeit small — at the wind farm scale (this is clearly shown in Stipa et al., 2023). Conversely, stability effects are only observable at the wind farm scale. For this reason, we refer to global blockage as the flow deceleration produced by the presence of stability above the ABL. When also individual turbine induction and its cumulative effect are considered, i.e. local blockage, the entirety of the upstream flow deceleration can be captured Devesse et al. (2023); Stipa et al. (2023). It is not clear to us what are the other physical mechanisms beyond gravity-wave-induced blockage the reviewer is referring to. We certainly acknowledge that global blockage will show a dependency on the wind farm geometry, wind shear, wind veer, and stability inside the ABL, but we argue that the physical mechanism still remains that described by, e.g. Devesse et al. (2023); Lanzilao and Meyers (2023); Stipa et al. (2023) and in the present paper.

Another important point is that focusing solely on global blockage only covers half of the underlying physics. In fact, the higher the blockage the more favorable the pressure gradient is inside the wind farm. As shown in Figure 11 of the revised paper, those conditions characterized by higher blockage are far from experiencing the lowest overall wind farm efficiency, a testament of the importance of unfavorable and favorable pressure gradients produced by stability both upstream and inside the wind farm.

To specifically address the reviewer's comment on flow confinement and AGWs contribution to blockage, we highlight the statement added at lines 605-609 of the track changes document. In particular, global blockage effects is always due to flow confinement, which is an alternative way of referring to the AGW-induced pressure gradient. The two are in fact uniquely related, as mathematically shown in Section 2.2 (this section has been heavily modified, please see the track changes document). To further expand on this, in the rigid lid cases, global blockage is generated in the exact same manner as in the full AGW solution, with the only difference being that flow confinement is restricted to that produced when $\eta = 0$. This implies that the flow is horizontally divergence free in the rigid lid case (i.e. on wall-parallel planes),

while continuity is satisfied on pliant surfaces defined by $\eta$ in the full AGW solution (i.e. curved surfaces, locally coincident with the vertical streamline displacement). Notably, both induce global blockage due to flow confinement or, alternatively, to stability effects above $H$, with the rigid lid being a limiting case for $\Delta\theta \to \infty$ and/or $\gamma \to \infty$. Additions to the revised manuscript regarding these aspect can be found at lines 328-333, 605-609 and 730-737 of the track changes document.

Finally, Figure 6 (which became Figure 7 in the revised manuscript) indicates that global blockage corresponding to $\eta = 0$ (rigid lid) yields the majority of the blockage observed when the flow confinement accounts for the AGW solution in the free atmosphere. In both cases, global blockage is produced by flow confinement.

To better elucidate the relation between fully neutral conditions (no stratification) in which blockage is only produced by turbine induction, the rigid lid condition which only considers the effect of $H$, and the full AGW solution, which also considers the effect of $\Delta\theta$ and $\gamma$, we have calculated the non-local, wake and wind farm efficiencies for each of these cases. These are defined by Lanzilao and Meyers (2023) and are reported in Section 4.2 of the revised manuscript. Besides enhancing our understanding of detrimental (global blockage) and beneficial (turbine wake recovery) effects produced by stability, we arrive to the same conclusion of Section 4.2 of Bleeg and Montavon (2022), i.e. that the rigid lid approximation might overestimate wind farm power even more than fully neutral conditions, even though it captures some of the global blockage effects. This emphasizes the importance of modeling the entirety of free atmosphere stability effects and not only global blockage.

Rigid-lid approximation: The authors use the rigid-lid approximation throughout the manuscript; however, it is not clear what is the purpose of using such a simplified and unrealistic modeling approach. In Lines 83-85, the authors suggest the rigid-lid approximation may be useful for use in engineering parameterizations. What do the authors mean by engineering parameterizations? Also, the rigid-lid approximation is tested here neutral boundary layer flow, which is unrealistic compared to the atmospheric boundary layer. For example, Bleeg and Montavon (2022) show that neglecting the temperature stratification in the capping inversion and troposphere misrepresents the blockage effect.

By engineering parametrizations we refer to low-cost reduced order models such as the 3LM Allaerts and Meyers (2019) or the MSC model (Stipa et al., 2023). Reduced order models based on the rigid lid approximation are currently being used in industry tools to model global blockage effects. This aspect is also reported in Section 4.2 of Bleeg and Montavon (2022), where it is referred to as the symmetry plane method. The two things are in principle equivalent.

The last comment made by the reviewer implies that the basic idea of the approach described in the manuscript is not clear. To rectify this, a clarification about the purpose of investigating the rigid lid approximation has been added to the revised manuscript and can be found at lines 591-593 of the track changes document. In particular, the fact of imposing a certain height and displacement of the upper boundary in the LES is automatically equivalent to consider a certain inversion strength and free atmosphere lapse rate (see Section 2.2 of the revised manuscript, where this is explained using a simple analytical model). This means that the effect of stability above the ABL can be implicitly modeled within the ABL if the vertical boundary layer displacement $\eta$ corresponding to the specific conditions under investigation (wind farm geometry, and unperturbed velocity and potential temperature profiles) are known. This is because the heterogeneous pressure gradient produced by flow confinement due to $\eta$ and arising from the AGW solution in the free atmosphere have to be coincident. As a consequence, the rigid lid approximation is not equivalent to a case without thermal stratification, where the upper boundary should be placed ideally very far from the ground. Instead, it is a limiting solution corresponding to very large stratification above the ABL. The purpose of studying this approximation is to understand how it compares with current industry practice (i.e. fully neutral conditions) and with the full AGW solution. The same limiting solution has been studied by Bleeg and Montavon (2022) (cases 3 vs 5 and 3b vs 5b).

Line 153-154: Why are the wind farm and upper layer characterized by the same background velocity? This assumption virtually discards the effect from shear and the large gradients associated with the atmospheric surface layer.

The analytical model presented in Section 2.2 is only used to explain the unique relation that exists between the pressure $p^\star$ and the inversion displacement $\eta$. The assumption of constant velocity inside the boundary layer allows the original 3LM equations to be easily rewritten in terms of $\eta$ instead of $\eta_1$ and $\eta_2$ by summing up the continuity equations in the wind farm and upper layer. Doing the same within the original 3LM equations would only be possible in Fourier space and the conclusions would be more difficult to see. However, the generality of our reasoning can be readily proved by noticing that, once $\eta$ is known, Equation 10 is not required anymore and pressure can be obtained by solving Equation 9.

Figure 2: The divergent color map is not centered at 0, making it very difficult to distinguish between positive and negative inversion displacements.

The reviewer's comment has been implemented in the revised manuscript.

Lines 299-300: I would argue that the AGW-modeled and AGW-resolved approaches do not predict almost the same pressure perturbation for the subcritical case (Figure 3a). Differences in the pressure perturbation field between the AGW resolved and modeled approaches are at least on the order of 10% upstream of the wind farm.

We agree with the reviewer's comment and the paragraph has been rephrased.

Lines 307-312: The differences upstream of the wind farm are just as large (or larger) than the differences at the domain outflow. However, the hypothesis presented by the authors does not address these differences. The flow upstream of the wind farm is outside and downstream of the fringe region and these differences are still large.

We agree with the reviewer's comment and added additional explanation in the revised manuscript. In particular, the following considerations can be made. The AGW-modeled and MSC model feature the exact same $\eta$, but a different level of fidelity inside the boundary layer. Hence, the same $\eta$ does not lead to identical pressure and velocity perturbations. Conversely, the AGW-modeled and AGW-resolved cases use the same model inside the ABL, but $\eta$ is slightly different, as it comes from the MSC model in the former and it is resolved in the latter. As a consequence, mass and momentum conservation show some differences in the perturbation velocity and pressure. This aspects have been added to the revised manuscript (see lines 518-529 of the track changes document).

Lines 344-345: The authors conclude that flow confinement is responsible for blockage to a lesser extent than gravity waves. However, Figure 6 clearly shows that the velocity deceleration with gravity waves is less than twice as large as the deceleration in the rigid-lid simulations. Thus, it seems flow deceleration from gravity wave-induced pressure gradients is not the main cause for blockage in these simulations. Also, I would argue that flow confinement is not the only cause for blockage in the rigid-lid case.

We realized that the way the sentence was written was misleading. In particular, blockage is in both cases given by flow confinement effects. In the AGW-modeled cases, the flow confinement is produced by an inversion displacement calculated with linear theory using the MSC model (thus accounting for the full AGW solution). In the rigid lid case, the value of $\eta$ has been set to zero according to an infinitely high stability above the boundary layer (which is an approximation). Figure 6 (corresponding to Figure 7 in the revised manuscript), shows that assuming an infinitely strong stability accounts for the majority of the

global blockage effect observed when modeling also the inversion displacement. The sentence has been rephrased according to these considerations (see lines 605-609 of the track changes document).

The authors mention that the LES domain should extend to one or more wavelengths in each direction (Line 113). However, extending the LES above 10-12 km in the atmosphere means you are performing simulations above the tropopause, where the temperature stratification is very different from the constant lapse rate assumed within the troposphere. Is gravity wave propagation sensitive to having multiple thermally stratified layers like in the atmosphere compared to a single constant lapse rate? This might be out of the scope of the paper but is something to consider.

This is correct. While the validity of the Boussinesq approximation for such tall domains has been demonstrated (see "Response to Reviewer 2" from `https://wes.copernicus.org/preprints/wes-2023-40/wes-2023-40-AR2.pdf`), the assumption of linear lapse rate might be somewhat strong for certain LES setups where $\lambda_z$ is very large. However, although undoubtedly worthwhile to be kept in mind for the future, this aspect has not been addressed yet in the context of wind farm LES. The only study that looked at non-uniform lapse rate in the free atmosphere has been performed by Devesse et al. (2022), who extended the 3LM to model these types of conditions. The authors state that a non-uniform lapse rate can play a big role in some cases. However, the 3LM model is based on linear theory and cannot account for other important physical phenomena such as gravity wave break-up and non-linear interaction.

**References**

Allaerts, D. and Meyers, J.: Sensitivity and feedback of wind-farm-induced gravity waves, Journal of Fluid Mechanics, 862, 990–1028, https://doi.org/10.1017/jfm.2018.969, 2019.

Bleeg, J. and Montavon, C.: Blockage effects in a single row of wind turbines, Journal of Physics: Conference Series, 2265, 022 001, https://doi.org/10.1088/1742-6596/2265/2/022001, 2022.

Devesse, K., Lanzilao, L., Jamaer, S., van Lipzig, N., and Meyers, J.: Including realistic upper atmospheres in a wind-farm gravity-wave model, Wind Energy Science, 7, 1367–1382, https://doi.org/10.5194/wes-7-1367-2022, 2022.

Devesse, K., Lanzilao, L., and Meyers, J.: A meso-micro atmospheric perturbation model for wind farm blockage, Submitted to Wind Energy Science Journal, URL `https://arxiv.org/abs/2310.18748`, 2023.

Lanzilao, L. and Meyers, J.: A parametric large-eddy simulation study of wind-farm blockage and gravity waves in conventionally neutral boundary layers, 2023.

Stipa, S., Ajay, A., Allaerts, D., and Brinkerhoff, J.: The Multi-Scale Coupled Model: a New Framework Capturing Wind Farm-Atmosphere Interaction and Global Blockage Effects, Wind Energy Science Discussions, 2023, 1–44, https://doi.org/10.5194/wes-2023-75, 2023.

---

## Referee Report (RR1)

The revised manuscript is well presented and well structured. The authors response covers all doubts and questions raised, and appropriate changes have been applied to the manuscript. However, I still have some minor scientific and technical comments which you can find here below.

Scientific comments/questions

1. Line 280: the authors mention that "The cells in between will be displaced between zero and η depending on their distance from the wall". However, to my understanding, the vertical cells are not only displaced but also stretched to account for the changes in the vertical domain height (which depend on η). Is this correct?

2. Line 444-461: I would suggest expressing the differences in pressure between the various cases in percentage since stating "small pressure deviation" or "large differences" can be subjective. I would suggest to apply this change throughout the manuscript, where possible.

3. Line 470: note that a difference of 5% in terms of velocity causes a difference in power output of 15%, which is substantial.

4. Figure 6: it could be useful to also include the relative error, as done for Figure 5.

5. I would suggest changing the title of Section 4.2. Here, the authors do not show results about the rigid lid approximation but rather compare these results to the ones obtained under different types of thermal stratification.

6. Figure 9: it is shown that the rigid lid case has a higher wind farm efficiency than the truly neutral case. Which mechanism is responsible for this behaviour? Could it be that the flow speed-up over the farm (if present) in the rigid lid case enhances vertical mixing and therefore wake recovery?

7. A somewhat similar analysis to the one presented in Figure 9/10/11 has been performed by Allaerts and Meyers (2019) and Lanzilao and Meyers (2024). It would be interesting if the authors could relate their findings to the ones described in the articles mentioned above, when possible.

Technical comments

1. Line 45: error in reference style (\cite{} -> \citep{})

2. Line 304: untested -> has not been tested

3. Line 314: remove "used to compute η"

4. Line 656: it could be more intuitive to express the time in hours instead of seconds

5. Line 665: replace with "the geostrophic wind is not know a priori and it has to be retrieved by.."

6. Table 3: I would suggest replacing N1 and N2 with subcritical and supercritical, respectively.

7. Table A1: in the caption, change u\ast to u_\ast

---

## Author Response (AR2)

**University of British Columbia**

**UBCO-UL NSERC Alliance Grant "Reduced-Order Models of Wind Farm Induction and Far-Field Wake Recovery"**

**Response to Reviewer 1**

Exec. S. Stipa - May 7, 2024

We would like to thank the reviewer for the time dedicated to revising the paper. We proceed with answering and clarifying, where possible, their comments.

Our response, denoted in black, is shown below, while the reviewer's comments are denoted in blue. Please refer to the track changes document for a detailed overview of the changes made to the manuscript.

Line 280: the authors mention that "The cells in between will be displaced between zero and $\eta$ depending on their distance from the wall". However, to my understanding, the vertical cells are not only displaced but also stretched to account for the changes in the vertical domain height (which depend on $\eta$). Is this correct?

The reviewer is correct in pointing out that this sentence is still not clear enough. The sentence has been corrected, highlighting that indeed is not the mesh cells which are displaced, but rather the mesh points. As a consequence, cells are stretched vertically with respect to the starting configuration where the mesh is initially cartesian.

Line 444-461: I would suggest expressing the differences in pressure between the various cases in percentage since stating "small pressure deviation" or "large differences" can be subjective. I would suggest to apply this change throughout the manuscript, where possible.

We implemented this comment throughout the manuscript, removing the instances of small/smaller and large/larger where possible. Regarding the specific line pointed out by the reviewer, we emphasize that providing the difference in percent of the pressure calculated from the AGW-resolved simulations is not a good metric, as deltas would go to infinity when the pressure perturbation values cross the zero perturbation line (due to a division by zero). Instead, we added the pressure delta in percent of $\rho u_{\mathrm{ref}}^2$, which is also consistent with the plots.

Line 470: note that a difference of 5% in terms of velocity causes a difference in power output of 15%, which is substantial.

We agree with the reviewer. We also highlight that our percentages are calculated relative to the local velocity, instead of $u_{\mathrm{ref}}$. This means that a constant difference in velocity between the two plots would see an increase in relative error inside the wind farm, where the local velocity is lower. We are comfortable with keeping this metric in the manuscript, but we are also open to change it if the reviewer thinks that dividing by $u_{\mathrm{ref}}$ instead of $u(x)$ would make the plots more clear.
On a different note, differences in velocity around 5% (and more) can be also due to the different inflow used between the AGW-resolved and AGW-modeled simulations, as shown in Appendix B of the manuscript, so this does not mean that our approach implies a difference in power of 15%. Specifically, while the exact absolute value of velocity — and hence power — depends largely on the specific inflow used, our objective here is to show that our approach can capture the effect of AGW on the velocity gradients around the wind farm and on the row-by-row thrust and power distributions. These can be observed from Fig. 5 and 6, respectively, where we believe that the validity of the proposed approach can be appreciated.

Figure 6: it could be useful to also include the relative error, as done for Figure 5.

As mentioned above, we believe that only focusing on the relative error between the AGW-modeled and AGW-resolved approaches can be misleading, as differences also contain the effect of using a different inflow. For this reason, we do not agree with the reviewer that adding the relative difference would add clarity to our results. Perhaps the relative difference in Fig. 5 caused some confusion. However, we include the relative power difference in Fig. 1 of this response, calculated as $(P_{AGWM}^t - P_{AGWR}^t)/P_{AGWR}^t$ (where $t$ is the row ID), for the subcritical and supercritical cases.

[Figure]

**Figure 1.** Comparison of row-averaged error on power for the subcritical (top) and super-critical (bottom) cases. Time averaging is performed as described in Appendix B of the manuscript.

I would suggest changing the title of Section 4.2. Here, the authors do not show results about the rigid lid approximation but rather compare these results to the ones obtained under different types of thermal stratification.

We agree with the reviewer, the title has been changed to "Implications of the Rigid Lid Approximation".

Figure 9: it is shown that the rigid lid case has a higher wind farm efficiency than the truly neutral case. Which mechanism is responsible for this behavior? Could it be that the flow speed-up over the farm (if present) in the rigid lid case enhances vertical mixing and therefore wake recovery?.

While we highlight that we did not run any truly neutral LES and so the data to precisely answer the question is not available to us, we hypothesize that the higher wind farm efficiency in the rigid lid case with respect to the fully neutral case is primarily due to the lid-induced pressure gradient around the wind farm, where the favorable gradient inside overcomes the unfavorable gradient upstream. Moreover, as momentum fluxes due to vertical mixing are mainly produced by the $\partial \overline{u'w'}/\partial z$ term in the momentum equation, where the production of $\overline{u'w'}$ indeed increases with increasing shear, the hypothesis of the reviewer might also be a factor. Nevertheless, in our experience the acceleration above the wind farm in a capped boundary layer, which arises due to continuity arguments, does not necessarily translate in more efficient vertical mixing. An example is a stably stratified boundary layer, where $\overline{u'w'}$ are quickly suppressed by buoyancy after the wind farm, causing the flow to remain accelerated in the upper layer as little to no momentum is removed by turbulence.
However, we emphasize that the exact dynamics of the shear stress evolution inside a finite wind farm under a truly neutral boundary layer has never been compared to a capped CNBL to date, representing an important topic to investigate in the future. For instance, although this question was not answered in their paper, the data from the parametric study performed by Lanzilao and Meyers (2024) can be used to address this topic.

A somewhat similar analysis to the one presented in Figure 9/10/11 has been performed by Allaerts and Meyers (2019) and Lanzilao and Meyers (2024). It would be interesting if the authors could relate their findings to the ones described in the articles mentioned above, when possible.

We expanded a bit this section with some comments that relate our findings to those proposed by the reviewer. There is one fundamental difference between the parametric study conducted by Allaerts and

Meyers (2019) and ours. Since the coupling between the wake model and the atmospheric perturbation model in the MSC model is local, it can account for both adverse and favorable AGW-induced pressure gradient effects. Conversely, the 3LM used by Allaerts and Meyers (2019) only accounts for blockage effects. As a consequence, these authors found that those conditions characterized by a high blockage are also those where the wind farm produces less. This is different from our findings, where more blockage also means more beneficial pressure gradient, leading to a higher wake recovery and a higher overall power production by the wind farm. This is true for the range of $H/H_1$ where the MSC is applicable, but it should be pointed out that, in general, the dominance of adverse over favorable pressure gradient also depends on the ABL height, as shown by Lanzilao and Meyers (2024). In particular, these authors show that, for very low ABL heights ($H \approx H_1$), $\eta_{\mathrm{nnl}}$ correlates well with $\eta_{\mathrm{tot}}$, even though $\eta_{\mathrm{w}}$ increases. In this case, because wind turbines span the entire ABL height, the momentum loss cannot be replenished by vertical turbulent fluxes, leaving only the favorable pressure gradient to promote wake recovery.

Line 45: error in reference style (\cite -> \citep).

Rephrased and corrected.

Line 304: untested -> has not been tested.

Corrected.

Line 314: remove "used to compute $\eta$".

Rephrased.

Line 656: it could be more intuitive to express the time in hours instead of seconds.

We prefer to leave the time in seconds, as this is the standard SI and it is the unit of measure used in simulation codes.

Line 665: replace with "the geostrophic wind is not know a priori and it has to be retrieved by.."

It is not clear to us where the modification should be implemented.

Table 3: I would suggest replacing N1 and N2 with subcritical and supercritical, respectively.

Corrected.

Table A1: in the caption, change u\ast to u_\ast

Corrected all instances of $u_*$ to $u^*$ throughout the paper for consistency.

**References**

Allaerts, D. and Meyers, J.: Sensitivity and feedback of wind-farm-induced gravity waves, Journal of Fluid Mechanics, 862, 990–1028, https://doi.org/10.1017/jfm.2018.969, 2019.

Lanzilao, L. and Meyers, J.: A parametric large-eddy simulation study of wind-farm blockage and gravity

waves in conventionally neutral boundary layers, Journal of Fluid Mechanics, 979, A54, https://doi.org/10.1017/jfm.2023.1088, 2024.